# Atmospheric moisture supersaturation in the near-surface atmosphere at Dome C, antarctic plateau

Christophe Genthon[1], Luc Piard[1], Etienne Vignon[2], Jean-Baptiste Madeleine[3], Mathieu Casado[4],

Hubert Gallée[1]

[1] CNRS, LGGE, Grenoble, France

[2] Université Grenoble Alpes, LGGE, Grenoble, France

[3] LMD – IPSL, UP6 – CNRS, Paris, France

[4] LSCE – IPSL, CEA-CNRS-UVSQ-U. paris-Saclay, Gif-sur-Yvette, France

*Keywords:*

*Antarctic plateau*

*Atmospheric supersaturation*

*Hygrometer*

*In situ measurement*

*Cold microphysics*

*Atmospheric modeling*

*v061216*

**Abstract.** Supersaturation in the naturally occurs at the top of the troposphere where cirrus clouds

form, but is comparatively unusual near the surface where the air is generally warmer and laden

with liquid and/or ice condensation nuclei. One exception is the surface of the high antarctic

plateau. One year of atmospheric moisture measurement at the surface of Dome C on the East

Antarctic plateau is presented. The measurements are obtained using commercial hygrometry

sensors modified to allow air sampling without affecting the moisture content even in case of

supersaturation.  Supersaturation is found to be very frequent. Common unadapted hygrometry

sensors generally fail to report supersaturation, and most reports of atmospheric moisture on the

antarctic plateau are thus likely biased low. The measurements are compared with results from 2

models implementing cold microphysics parametrizations: the European Center for Medium-range

Weather Forecasts through its operational analyses, and the Model Atmosphérique  Régional. As in

the observations, supersaturation is frequent in the models but the statistical distribution differs both

between models and observations and between the 2 models, leaving much room for model

improvement. This is unlikely to strongly affect estimations of surface sublimation because

supersaturations are more frequent as temperature is lower, and moisture quantities and thus water

fluxes are small anyway. Ignoring supersaturation may be a more serious issue when considering

water isotopes, a tracer of phase change and temperature, largely used to reconstruct past climates

and environments from ice cores. Because  observations are easier in the surface atmosphere, longer

and more continuous in situ observation series of atmospheric supersaturation can be obtained than

higher in the atmosphere to test parameterizations of cold microphysics, such as those used in the

formation of high altitude cirrus clouds in meteorological and climate models.

**1. Introduction**


Ice supersaturation is frequently found in the upper troposphere  [Spichtinger et al. 2003] and

specific cloud microphysics parameterizations are developed to represent this process in

meteorological and climate models. These models need to be validated against observations to

reproduce cirrus and other clouds including contrails which develop at altitudes where

supersaturation occurs (e.g. Rädel and Shine [2010]). Radiosondes provide snapshot information but obtaining in situ observation timeseries to comprehensively calibrate and validate such parameterizations is a challenge because it requires flying and operating instruments on high altitude aircrafts or balloons. Sampling supersaturated air parcels without affecting the air moisture content is also a challenge, as the excess moisture with respect to saturation tends to condense on any surfaces including those of the sampling device and the sensor itself. There are thus not many in situ observations available to characterize and quantify natural supersaturations and their evolution in time, and evaluate and validate microphysics parameterizations in such conditions.

While they are frequent at high altitude, ice supersaturations do not generally occur in the surface atmosphere, where operating instruments is obviously much easier. Atmospheric conditions close to those occurring at the tropopause are however found at the surface of the antarctic ice sheet both in terms of temperature and humidity levels. Because of the distance from the nearest coasts and the high elevation, the antarctic plateau is also particularly secluded from sources of aerosols. This is the most likely place on Earth to observe frequent and large ice supersaturation in the near surface atmosphere. For instance, Schwerdtfeger [1970] reports on observations of relative humidity with respect to ice exceeding 120% at Vostok station in the heart of Antarctica.

The possibility of surface atmospheric supersaturation on the antarctic plateau raises a potential issue, that of the relative contribution of the different terms of the surface mass balance of the antarctic ice sheet. The terms are precipitation (positive for the surface) and evaporation/sublimation (negative or positive), and possibly blowing snow (positive or negative as blown snow redeposits, but generally negative because of enhanced snow evaporation, e.g. Barral et al. [2014]). Melting and runoff do not occur on the antarctic plateau and can be excluded. The net surface mass balance, observed using glaciological methods, is very small on the antarctic plateau.

It is typically a few cm water equivalent per year [Arthern et al., 2006]: the antarctic plateau is one

of the driest places on Earth. This is because it is so cold, and thermodynamics imply that the

various terms of the surface mass balance are bound to be correspondingly small. Because they are

so small, and because of a harsh environment, the direct determination of precipitation and

evaporation/sublimation on the antarctic plateau is not conclusive. Their relative contribution to the

surface mass balance of the antarctic plateau is still poorly quantified, using indirect approaches

[Frezzotti et al., 2004]. In most places on continents, precipitation largely dominates. This is not

necessarily the case on the antarctic plateau. In particular, if atmospheric supersaturation occurs

near the surface, then moisture concentration is likely larger in the surface atmosphere than at the

snow surface and the turbulent moisture flux is thus directed towards the surface (surface

condensation). Unlike most other regions of the Earth, this turbulent flux could contribute positively

to the surface water budget and thus, here, on the surface mass balance.

Another potential issue with ice supersaturation on the antarctic plateau is that of the impact on the

water isotopic composition of snow. Supersaturation leads to kinetic fractionation of the stable

isotopic composition of water when it condenses. Since the 1980's [Jouzel et al. 1987], the longest

ice core records of past climate and environment are obtained from drilling operations on the

antarctic plateau. Past atmospheric temperatures are deduced from the variations of the

concentration of stable water isotopes along the core. Variations in supersaturation levels may

impact kinetic fractionation and thus on this reconstruction. Supersaturations thus involve not only

meteorological (clouds, precipitation, surface evaporation / sublimation) but also climate and

paleoclimate reconstruction issues. It is therefore important to measure and assess supersaturations

on the Antarctic plateau.

However, as already mentioned, measuring atmospheric supersaturation is a challenge because

sampling a supersaturated air mass can affect its moisture content. Schwerdtfeger [1970] expresses

concerns about the reliability of reports of supersaturation at Vostok station. On the other hand,

many reports of relative humidity with respect to ice (RHi) on the antarctic plateau reach but seem

to be capped at 100% [King et al., 1999]. Genthon et al. [2013] compare RHi observed at Dome C

on the antarctic plateau using conventional solid state sensors with results from the ECMWF

(European Center for Medium-range Weather forecasts) meteorological analyses and from the MAR

(Modèle Atmosphérique Régional) meteorological model. In both models, cold microphysics

parameterizations are used which, depending on local conditions, allow for supersaturations

(section 4). More often than not, when ~100% RHi is observed at Dome C with conventional

instruments (not adapted to sample supersaturation), both models produce significant

supersaturation, occasionally reaching more than 150% [Genthon et al., 2013]. The cold

microphysics parameterizations differ in the 2 models (see section 4), and other aspects such as the

vertical resolution also differ. If both models produce significant supersaturations, they do not

quantitatively agree as to the amplitude of the supersaturations.

To verify such model results, to decide between and to improve he models, using direct is situ

measurements, instruments must be designed and/or adapted so as to bring the air mass to the

moisture sensor without affecting its moisture content. This can be done by warming the air above

its condensation temperature before ushering it to the sensor. Here, after the present general

introduction (Section 1) , Section 2 presents 2 instruments which are adapted from commercial

sensors to perform in very cold conditions and to enable the measurement of atmospheric

supersaturation at Dome C. The measurement site and deployment are also described in Section 2,

and previous atmospheric humidity reports from this site are revisited. In Section 3, results from the

conventional instruments are compared with the reports by the 2 adapted instruments and shown to

fail. A 1-year climatology of atmospheric moisture at Dome C from the adapted instruments is

presented, first for summer when both adapted instruments work well but not the unadapted one,

then for a full year when instrumental limits occur in the coldest and driest periods and are

discussed. The impact of the supersaturations on the turbulent exchange at the surface is calculated

and shown to be minor. In section 4, simulation results from recent versions of the 2 atmospheric

models discussed in Genthon et al. [2013]  are shown to agree with the observation of frequent

occurrences of supersaturation at all time in the year including in summer. It is also shown that

details of the climatology and the statistics of occurrence of supersaturation differ between the

models and the observations and between the 2 models.  Section 5 discusses the results, issues

related to limited ability of models to properly account for supersaturation, including potential

consequences for the record of isotopic signals in the ice, and finally concludes the paper.


**2. Measurement site, instruments and observation methods**

Dome C (Figure 1) is one of the main topographic domes on the east antarctic plateau. Since

2005, the summit of the dome (75° 06' S, 123° 20' E, 3233 m a.s.l.) has hosted a permanently

manned station, Concordia, jointly operated by the French and Italian polar institutes (IPEV and

PNRA). One of the first Antarctic Meteorological Research Center automatic weather station

(AMRC AWS, https://amrc.ssec.wisc.edu/) deployed in Antarctica, back in the 1980s, was at Dome

C. When the actual location of the summit of the dome was later more accurately determined using

satellite and aircraft radar altimeters in the 1990s, the AWS was moved about 50 km to its present

position. This induced a 30 m rise and correspondingly slight mean surface pressure change but

otherwise little impacted on the series consistency because the local environment is very

homogeneous. The AWS provides one of the longest quasi-continuous meteorological reporting on

the high antarctic plateau. The station measures pressure, temperature and wind, but not moisture.

Additional meteorological reports are available since the construction of Concordia station,

including another AWS closer to the station and a daily radiosonde. Both the new station and the

radiosondes report atmospheric humidity using solid state film capacitive sensors [Kämpfer et al.,

2013]. In early 2008, a system to vertically profile the lower part of the atmosphere was deployed

along an ~45 m high tower. Temperature, wind and moisture are measured, the latter again using

solid state film capacitive sensors. This profiling system is fully described in Genthon et al. [2010],

Genthon et al. [2011] and Genthon et al. [2013].

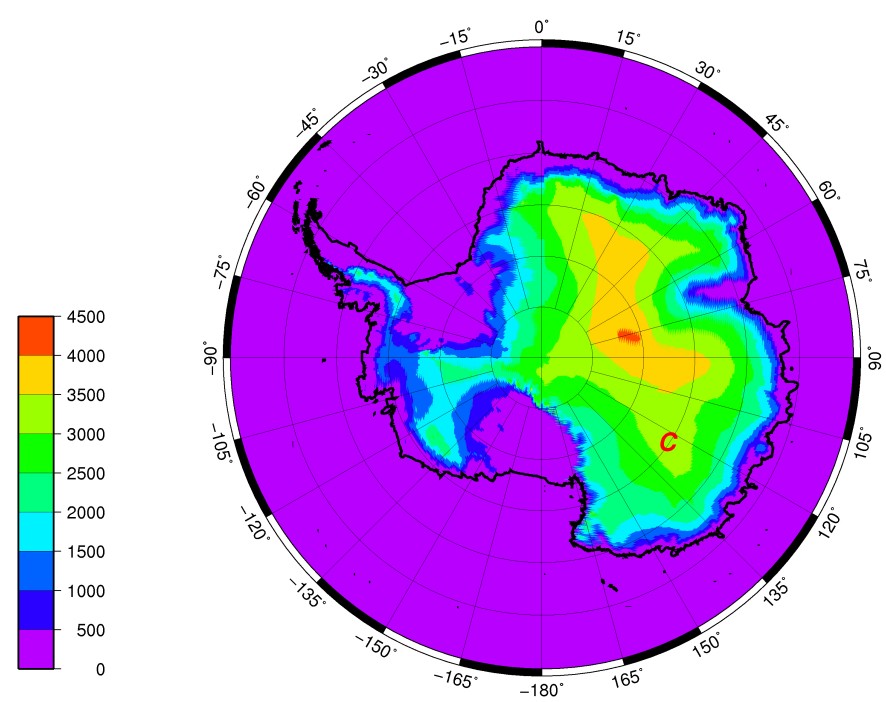

*Figure 1: Topographic map of Antarctica, showing the location of Dome C (red C). Elevation*

*(color scale) is in m.*


From the tower measurements, Genthon et al. [2013] evaluate and compare 2 contrasting

years, 2009 and 2010, respectively the warmest and coldest in a 10-year period. They report

measuring humidity up to ~100% with respect to ice but also observing frequent frost deposition, a

hint that supersaturation occurs but is missed by standard hygrometers without an adaption.

Occurrences of supersaturation are further supported by a comparison with models that implement cold microphysics parameterizations: the models often simulate supersaturation when the hygrometers hit the 100% RHi ceiling. That raw solid state hygrometers cannot measure supersaturation is understandable: a supersaturated air mass will deposit its excess moisture on any hard surface that serves as a condensation surface. The hygrometer body itself will condense the

excess moisture before it can be measured. One way to overcome this problem is to aspirate and warm the air above its thermodynamic saturation temperature at the intake.

There are several techniques to measure atmospheric moisture [Kämpfer, 2013]. The traditional wet bulb thermometer is not very practical, particularly when measuring well below

freezing temperature. The dew-point hygrometer provides a direct physical measure of the saturation temperature. This is done by progressively cooling from ambient temperature a surface until atmospheric moisture condensation is detected on the surface. The cooled surface is generally a mirror and condensation is optically detected when the reflection of a light beam is observed to be diffused and diffracted. The device also works below freezing temperature but should then be

referred to as a frost-point hygrometer [King and Anderson, 1999]. Dew and frost-point hygrometers are accurate but bulky, complex and expensive. They require significant amounts of energy, and they have moving parts because the mirror must be periodically cleaned. They are thus comparatively prone to disfunction and failures, and they cannot be used in remote unattended places or in radiosondes. On the other hand, they mechanically aspirate air to the sensing mirror,

and if the aspiration intake is heated significantly above ambient temperature (such as in King and Anderson [1999]), the measured air is sampled without affecting its moisture content even if supersaturated. Some commercial instruments ensure this such as the Meteolabor VTP6 Thygan described below.

190        In the 1970s, Vaisala Oy (Finland) developed a very different, very compact humidity sensor, the Humicap thin film capacitive sensor[1]. The dielectric properties and capacitance of a polymer film vary with the relative humidity of the ambient air. Although the physical processes for dependence have been described (e.g. Anderson [1994]), the relationship between capacitance and atmospheric moisture is an empirical one. The sensor needs calibration and a small but significant

uncertainty affects the measurement. The uncertainty increases as temperature decreases. On the other hand, the Humicap is convenient, very compact, comparatively inexpensive, robust, its use can be automated and deployed even in remote places and on radiosondes. It is thus currently widely used for such purposes. Thin film capacitive sensors are used in all automatic weather stations in Antarctica that report moisture as well as on the 45-m profiling system at Dome C

mentioned above, in the latter case bundled in Vaisala HMP155 thermometers – hygrometers (thermohygrometers) [Genthon et al 2013]. According to the manufacturer, the uncertainty is +/- (1.4+0.032 of the reading in % in the -60°C to -40°C temperature range. It is smaller at warmer temperature and may be expected to be larger below -60°C. However, then, the absolute moisture content of the atmosphere is smaller  and absolute measurement errors are correspondingly smaller.


To tentatively confirm and quantify supersaturations at Dome C, both frost-point and thin film capacitive hygrometers were deployed at a height of 3 m and adapted as necessary to operate in the general Dome C conditions and to sample the air without altering its moisture content even when above saturation. In both systems, the hygrometer aspirated intake is heated so that the temperature

of the sampled air parcel is raised above condensation level and condensation is avoided. The frost-point hygrometer is a Meteolabor VTP6 Thygan chilled mirror instrument. It was selected because it is factory-designed to perform in cold temperatures and correspondingly low specific humidities.

---

1   http://www.vaisala.com/Vaisala%20Documents/Technology%20Descriptions/HUMICAP-Technology-description-B210781EN-C.pdf?utm_campaign=CEN-TIA-G-Humidity%20Nurturing%202015&utm_medium=email&utm_source=Eloqua&utm_content=CEN-TIA-G-HUMICAP%20Technology%20Promotion

According to the manufacturer the lowest measurable frost point temperature is -65°C. The fact that the air is heated at the intake (see below) does not improve the temperature range of the instrument as the actual limitation is due to the ability to cool the mirror to the condensation temperature. A -65°C temperature limit is not quite low enough to consistently operate at Dome C, where the surface atmospheric temperature can occasionally drop below -80°C. In addition, the sensor was found to begin and increasingly fail below -55°C rather than -65°C. However, data from the vertical profiling system show that from 2009 to 2015, the air temperature ~3 m above the snow surface was warmer than -55°C more than 50% of the time, and almost consistently (more than 99.5% of the time) warmer than -55°C during the local summer (Dec – Jan – Feb). Assuming near saturation, the instrument can nominally operate for a large fraction of the time at Dome C. For our application, the frost-point hygrometer (noted FP from now on) is hosted in a heated box so that the electronics and mechanics are not affected by the extreme cold temperatures in winter. By factory design, the outside air is aspirated inside the instrument through an intake protected by a heated hood which prevents frost deposition. This design is not modified, the intake and heated hood being simply made to protrude out of the heated box, to sample the outside air. This is the only part of the instrument outside the heated box and, because it is itself heated, loss of moisture along the way to the mirror is consistently prevented. Visual inspection confirms that even when frost deposition occurs on other instruments on the tower, no frost deposition is observed in the vicinity of the instrument intake. Each measurement cycle lasts 10 minutes: heating and defrosting the mirror from the previous measurement, cleaning, then cooling until frost point is reached. The sensor thus reports measurements of frost point temperature, and conversion to relative humidity, on a 10' time step basis. The manufacturer claims a very high accuracy: 0.1% expressed in term of relative humidity. Dew and frost point hygrometers are indeed often used to calibrate other types of hygrometers. Here the FP is used as a reference against which other sensors may be adjusted and are evaluated, at least down to temperatures where the FP performs well.

For the other type of hygrometer used here, the manufacturer (Vaisala Oy) guaranties its HMP155

sensor down to -80°C for the measurement of temperature, but only to -60°C for the measurement

of moisture. However, the main issue with colder temperatures for this instrument is that the time

response increases. Yet, unlike in a radiosonde for which the environment quickly varies during

ascent, variations are comparatively slow for fixed instruments and the operational limit is actually

much below -60°C [Genthon et al., 2013]. In addition, to avoid frost deposition and preserve the air

moisture content, for our application, the instrument aspirates the air through an inlet consistently

heated ~5°C above the ambient temperature (Figure 2). The ambient temperature itself is measured

by a separate PT100 platinum resistance thermometer in an unheated derivation of the system. A

comparison with the frost point hygrometer shows that this simple and low cost innovative design

succeeds to measure even highly supersaturated air. In addition, the fact that the air reaching the

hygrometer sensor is 5°C above ambient temperature correspondingly extends the actual nominal

temperature range of the instrument with respect to ambient temperature. The sensor reports relative

humidity. According to manufacturer, the accuracy in the low temperature range (-60° to -40°C) is

+/-1.4 % of the reading. Accuracy improves at warmer temperature, and may conversely be

expected to deteriorate for even colder temperatures. The temperature range -40° to -60°C is typical

at Dome C although temperatures as cold as -80°C and as warm as -15°C may be encountered.

Note, that in accordance with meteorological conventions, all sensors report relative humidity with

respect to liquid water rather than ice even when the air temperature is below 0°C. Goff-Gratch

formulas [Goff and Gratch, 1946] are commonly used in meteorology to convert between RH with

respect to liquid, water vapor partial pressure and RHi: they are also used here. Differences up to

20% have sometimes been reported with alternate formulaes in extremely cold temperature ranges.

However, because the formulae are used here to converted form RH with respect to liquid water to

partial pressure then to RHi, the inaccuracies partially compensate.

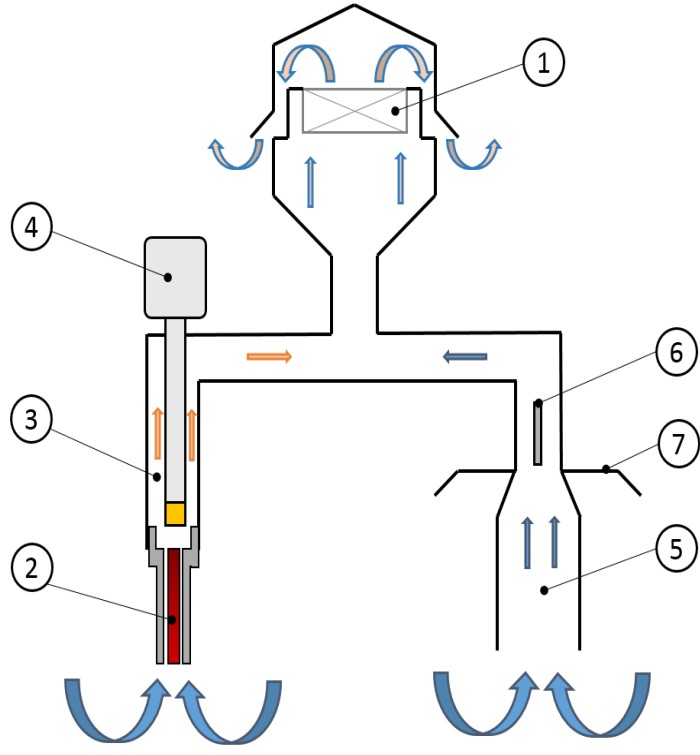

*Figure 2: Schematic drawing of the modified (HMPmod) hygrometer. The air is aspirated by the fan (1) and heated through an inlet (2). The  temperature and the moisture content of the heated air (3) is measured by the HMP155 (4). The ambient air  temperature (5) is measured by a separate PT100 (6) located in the unheated aspirated inlet shaded from sun radiation (7).*


A potential concern with the heated inlet approach is that, if there are airborne ice particles, they may be aspirated and evaporated in the heated section leading to spuriously elevated water vapor ratio in the sampled air, translating into supersaturation when RHi is recalculated against the ambient air temperature. Airborne ice particles may result from either blowing snow or occurrence

of a cloud. A signature of the bias would thus be that supersaturation magnitude and occurrence increases with wind speed and/or with downwelling infrared radiation. The reason is that blowing snow occurs if the wind is strong enough to erode and lift snow from the surface, while even a very

light cloud in such a cold and dry atmosphere induces a significant increase the IR emissivity and thus of downward IR (Gallée and Gorodetskaya [2000], Town et al. [2007]). In the observations to be presented next, the correlations are actually negative. RHi is consistently at or below 100% for wind speed above 8 m.s$^{-1}$ (figure 3), which is a typical speed for which blowing snow can trigger [Libois et al., 2014]. Stronger winds are generally associated with air masses originated from the coast and thus comparatively laden with aerosols preventing supersaturation. Supersaturations sharply increase rather than decrease in frequency and amplitude with downwelling IR below ~130 W.m$^{-2}$ characteristic of a clear sky (not shown). These results are fair signals that airborne ice particles are not likely to bias the measurements presented here. Caution with the reliability of the most extreme, and less frequent supersaturation events may nonetheless by recommendable.

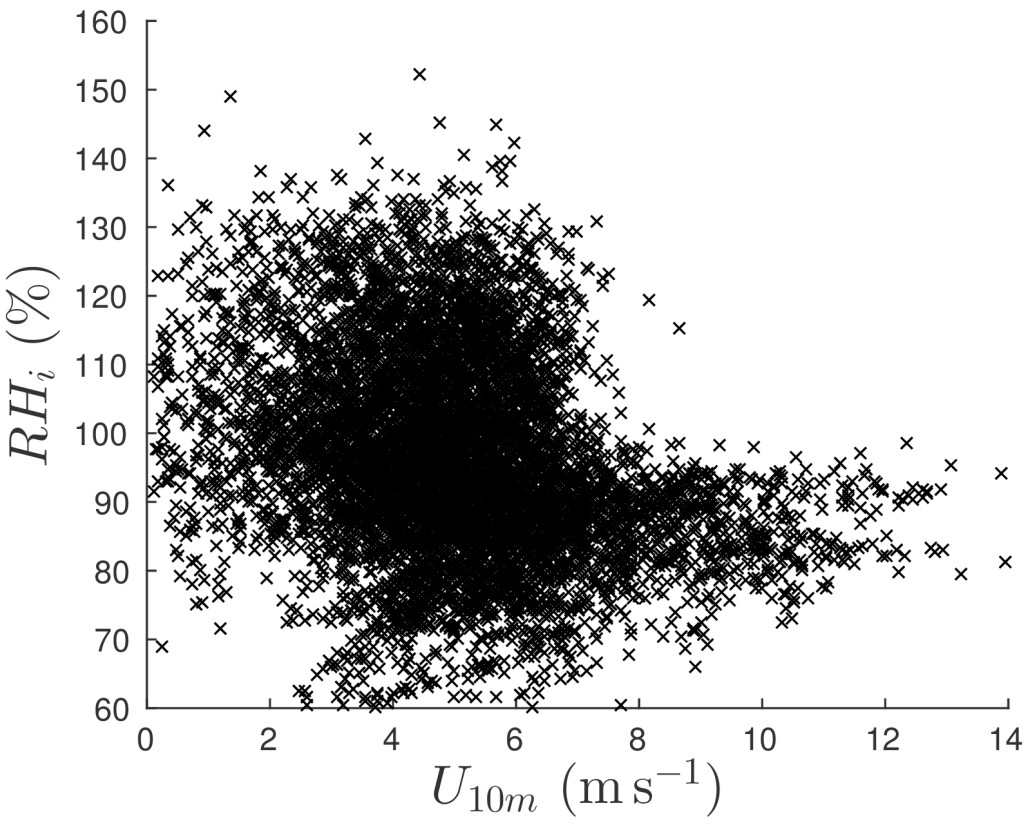

*Figure 3: Scatter plot of observed RHi between 60 and 160 % (from HMPmod) versus 10-m wind speed. All available half-hourly data in 2015 are plotted.*

The  2 adapted instruments are deployed side by side ~3 m above the snow surface on the ~45-m

tower. At the same level, hosted in an aspirated but unheated radiation shield (see Figure 1 of

Genthon et al. [2011]), an unmodified HMP155 allows for comparison with a traditional design –

and to exhibit biases of the latter. From now on, the original and modified HMP155 will be referred

to as "HMP" and "HMPmod", respectively. Table 1 lists the instruments and adaptations. The

various instruments performed over the duration of 2015 except for limited periods due to

datalogging failures or servicing in summer. The results are presented and analysed in the next

section and compared with models in section 4.


| Short name | Instrument / sensor | Housing |
|---|---|---|
| HMP | Vaisala HMP155 thermohymeter / thin film polymer hygrometer | Aspirated radiation shield |
| HMPmod | Modified Vaisala HMP155 thermohygrometer / thin film polymer hygrometer | Aspirated radiation shield + heated intake (Figure 2) |
| FP | Meteolabor Thygan VPT6 mirror frost-point hygrometer | Heated enclosure, heated intake |

*Table 1: List of hygrometers and adaptations. See text for details*

**3. Observation data and results**

**3.1. Summer**

Figure 4 displays the mean diurnal cycle of atmospheric moisture and temperature in January,

February and December 2015 according to the various instruments. During this period, the FP is

consistently running within its nominal manufacturer-stated temperature range and can serve as a

moisture measurement reference for the other instruments. The sun never completely sets at this

time of the year, however its changing elevation above the horizon induces a strong temperature

cycle near the surface (figure 4d). Here, "night" refers to the local hours during which sun elevation

is lower at Dome C and sets at lower latitudes, broadly the coldest half of the day. Figure 4a shows

the mean cycle of partial pressure of water vapor from FP. The numbers are low due to the cold

temperature: the water partial pressure ranges on average between ~15 Pa in the early morning and

slightly over 35 Pa in the early afternoon. This cycle demonstrates that surface evaporation occurs

during the day, followed by deposition at night, resulting in surface (3-m) atmospheric moisture

diurnally changing by a factor of more than 2. Figure 4b shows small differences and consistent

agreement between the HMPmod and FP instruments. Note here that HMPmod is slightly calibrated

for moisture report against the FP instrument for agreement in the early afternoon at the warmest of

the day. This calibration does not exceed manufacturer stated accuracy for HMP155 (Section 2).

The calibration proves robust and valid at all times during the day in this period. Results from

(unmodified) HMP significantly depart from those of the FP, and thus HMPmod instruments: the

agreement is good in the afternoon only, but quite poor the rest of the day and at night. Figure 4c

displays the calculated RHi for the 3 instruments, using the independent moisture measurements by

each instrument ,but all finally reported to one same atmospheric temperature, that of the

(unmodified) HMP. This is likely the most accurate estimation of temperature, i. e. the least likely

affected by radiation and other biases because it is unheated and most efficiently ventilated

[Genthon et al. ,2011]. Temperature differences of as much as 2°C are occasionally observed with

the other instruments in low wind conditions.

RHi differs markedly between the unmodified HMP and the 2 other instruments. The latter 2 both

report RHi significantly exceeding 100% while the unmodified instrument hardly reaches

saturation. All instruments agree well in the early afternoon at the warmest time of the day but HMP

disagrees at night. The FP and HMPmod instruments consistently agree with each other, including

when reporting averaged summer supersaturations reaching 120 % at night, confirming the high

levels of supersaturation hinted by Genthon et al. [2013] from models.


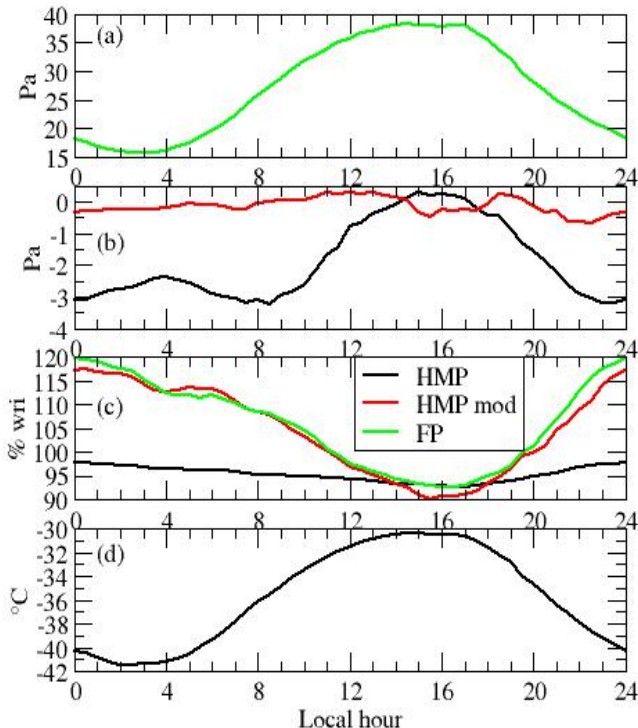

*Figure 4: Mean Dec-Jan-Feb diurnal cycle of: (a) water vapor partial pressure from FP instrument*

*; (b) difference with respect to FP of water vapor partial pressure from original (HMP, black) and*

*modified (HMPmod, red) thin film polymer sensors ; (c) RHi from the 3 instruments; (d) 3-m air*

*temperature.*

Figure 5 displays correlation plots of moisture reports from the unmodified (HMP) and modified

(HMPmod) thin film capacitive sensors with respect to FP in summer. The direct correlations

between water vapor pressures would be very high because humidity is largely controlled by

temperature. Plotting deviations to the saturation vapor pressure, rather than the vapor pressure

itself, removes much of the temperature codependence effect and concentrates on the relative ability

of the instruments to correctly measure moisture. The correlation between the regular HMP and FP

is good below saturation but is obviously very poor above since the HMP fails to capture

supersaturations. The correlation between HMPmod and FP reports is very high, above 0.97. The

regression constant (the intercept) is 0.1 but the standard error on the constant is larger than 0.1. The

linear regression is thus not statistically different from a 1/1 one.

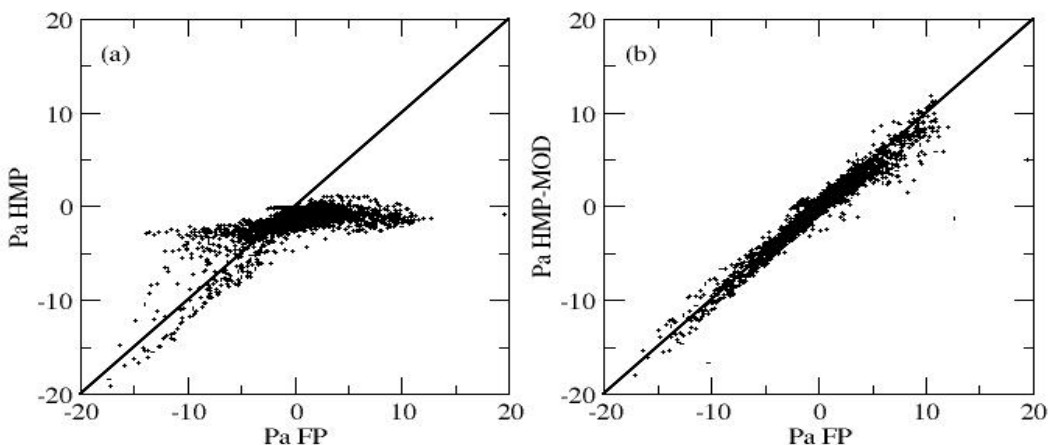

*Figure 5: regression of anomaly to saturation vapor pressure from HMP (a) and HMPmod (b)*

*instruments against FP.*

**3.2 Annual variations and statistics**

A strong diurnal cycle dominates the variability of atmospheric moisture in summer. The partial

pressure is maximum in the early afternoon while RHi peaks near local midnight (Figure 4) when it

occasionally reaches more than 150% (not shown). As the diurnal cycle variability progressively

vanishes and is replaced by synoptic variability in the colder months, RHi occasionally reaches

values above 200%. Figure 6 displays the distributions of observed RHi with the various

instruments, both limiting the range of RHi between 50 and 150% (more than 99% of all HMPmod

reports) and extending the range to 200%. A logarithmic RHi scale is used in the latter case because

with the linear scale the highest RHi values would almost merge with the axis and hardly be visible.

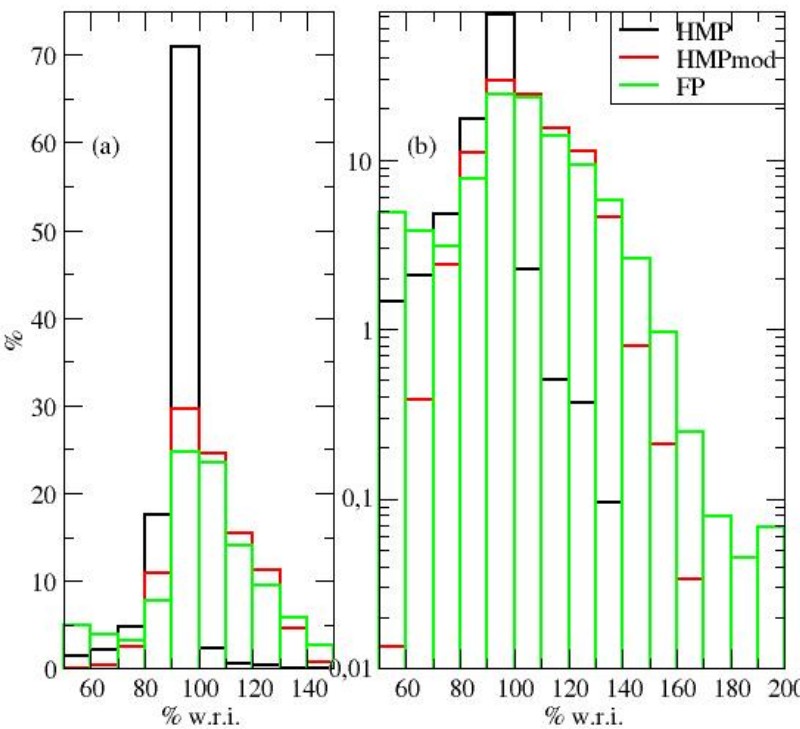

*Figure 6: Observed distributions of RHi in 2015 for cases of RHi between 50 and 150% with linear*

*vertical RHi scale (a) and between 50 and 200% and with logarithm vertical RHi scale (b).*


Although measurement uncertainties and uncertainties on conversions from relative humidity with

respect to liquid to RHi allow some occurrences above 100%, as expected, the reports from HMP

peak near and hardly exceed the 100% ceiling. More than 50% of all reports between 50% and

150% are above 100% for HMPmod and FP, with similarities of distribution for the 2 instruments.

There are differences between observations by even the 2 modified hygrometers though. In the 50-150% range, there are more quantitative differences between the HMPmod and FP below 100% than above. Both the HMP155 and frost point hygrometer lose accuracy and sensitivity as temperature is colder and/or water vapor partial pressure is less. Below -55°C, FP occasionally, and more and more frequently as temperature gets colder, reports unrealistically low moisture content. A

limit with the colder temperatures for this instrument is reported in section 2. Figure 7 displays the regressions of water vapor partial pressure differences with saturation, separately for partial pressure ranging between 2 and 5, 5 and 10, 10 and 20, and exceeding 20 Pa. The correlation deteriorates, and the regression line increasingly deviates from 1 to 1, as the moisture content decreases.


    Obviously, the smallest moisture partial pressures occur when the temperature is coldest. The instruments show their limits during the coldest periods of the winter. Figure 8 displays the annual cycle of monthly averaged temperature and RHi. HMP displays weak seasonal variability of RHi compared to the other instruments. On the other hand, FP displays extreme seasonal variability with

values reaching below 30% (beyond the plot scale on Figure 8) in winter. Such unrealistically low values, at odds with the other instruments, reflect instrument limitation with very low moisture content. Limiting the analysis to cases of partial pressure of moisture above 2 Pa (dashed curves on Figure 8) excludes significant portions of the coldest parts of the winter records. This is reflected by monthly winter temperature more than 20°C warmer (Figure 8a). The fact that HMPmod reports are

strongly increased suggests that this sensor also does not perform well at very low moisture levels. When restricting to above 2 Pa, both HMPmod and FP show strong seasonal variability with monthly mean RHi reaching 120% for HMPmod and exceeding 130% for FP. In both cases, the maximum monthly supersaturation is reached in early winter (April) and remains above 100% all year long, except in October for HMPmod when it is slightly below. Figure 9, same as Figure 6 but

for partial pressure of moisture above 2 Pa only, confirms that in the surface atmosphere of Dome

C, supersaturation is the norm rather than an exception.

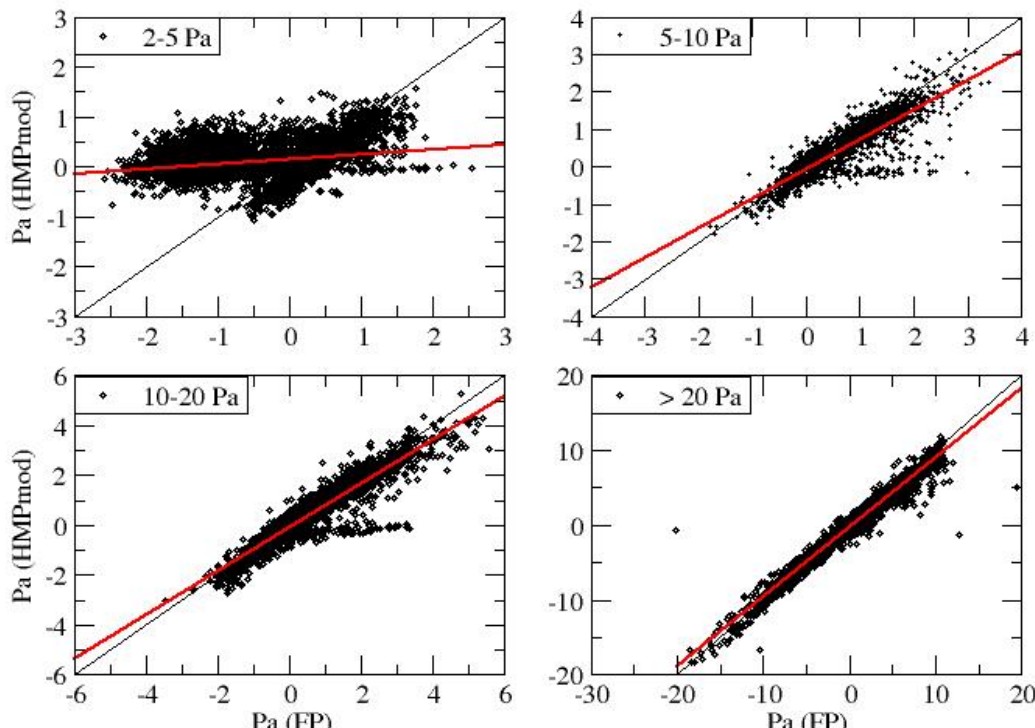

*Figure 7: Regressions of partial pressure difference with saturation from HMPmod against FP,*

*depending on partial pressure range as indicated on the upper left corner of each plot. The black*

*line is the 1$^{st}$ bisector, the red line shows the linear regression.*

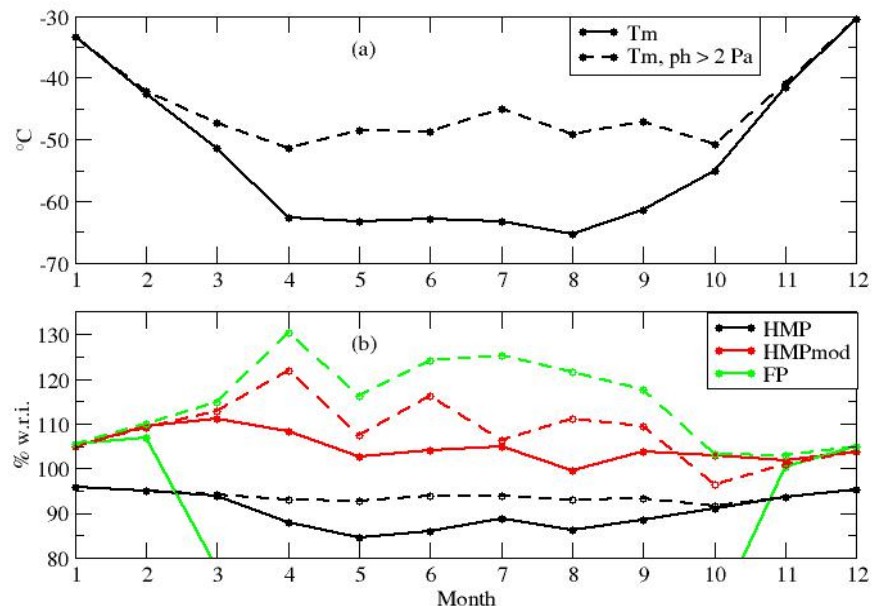

*Figure 8: Seasonal variability of monthly-mean temperature Tm (a) and RHi (b) for all reports*

*(solid lines) and reports with moisture partial pressure ph above 2 Pa only (dashed lines). With all*

*reports, the curve for FP reaches below 30%, well beyond the plot scale (green solid line).*

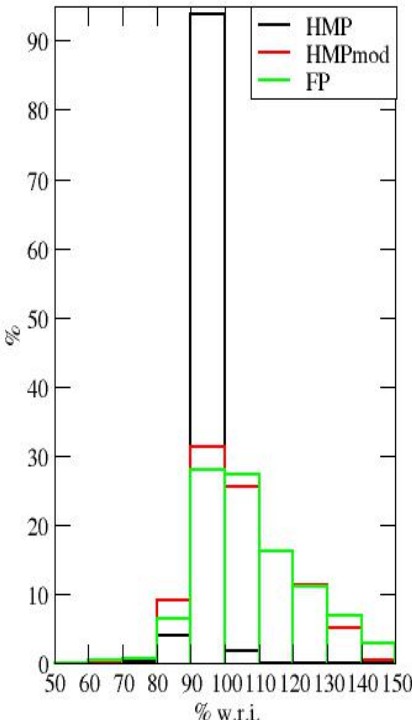

*Figure 9, same as Figure 6 but for moisture partial pressure above 2 Pa only.*

**3.3 Impact on surface sublimation calculations**

There are very few direct estimates of surface evaporation on the antarctic plateau. This is firstly

because eddy correlation techniques use delicate high frequency sampling instruments such as sonic

anemometers which are hard to operate and maintain at the required level of performance in the

extreme environment of the antarctic plateau. Moreover, due to the very low temperature, the water

vapor content is very small and moisture sensors are not both fast and sensitive enough for

measurement in such conditions. For instance, Van As et al. [2005] report that eddy correlation

measurements of latent heat flux were unsuccessful even in the summer at Kohnen station in

Antarctica ~3000 m above sea level. The authors thus resigned themselves to use bulk methods, a

most widely employed approach in Antarctica [Stearns and Weidner, 1993]. However, bulk methods

are equally affected by measurement biases such as underestimation of water vapor content due to failure to measure supersaturation. The magnitude of the error can be estimated at Dome C by

comparing bulk calculations using HMP and HMPmod water vapor reports.

The water vapor flux $E$ from the snow surface (subscript 's') to the atmosphere is calculated using bulk-transfer formulae :

$$E=\rho C_Q U(z)[q_s - q(z)] \qquad Equ.\ 1$$

where $\rho$ is the air density, $U(z)$ and $q(z)$ the wind speed and the specific humidity at the height $z$ in the atmospheric surface layer and $q_s$ the specific humidity at the surface, assuming saturation with respect to ice at the snow surface temperature. Here the wind speed and specific humidity are

measured at z=~3m above the surface, and the snow surface temperature is obtained from measurement of the upwelling infrared radiation [Vignon et al., 2016] considering a snow emissivity of 0.99 [Brun et al., 2011]. $C_Q$ is a bulk transfer coefficient which is written :

$$C_Q=\kappa^2\ [ln(z/z_0)-\psi_m(z/L)]^{-1}\ [ln(z/z_{0q})-\psi_q(z/L)]^{-1} \qquad Equ.\ 2$$


where $\kappa$ is the Von Kármán's constant, $z_0$ and $z_{0q}$ the roughness lengths for momentum and water vapor respectively and $\psi_m$ and $\psi_q$ are the corresponding surface-layer similarity stability functions. Stability functions depend solely on the dimensionless height $z/L$, where $L$ is the Monin-Obukhov length [Vignon et al. [2016], Stull [1990]. The same 4 function schemes taken for stable conditions

in Vignon et al. [2016] are tested here, and the functions from Hogström [1996] are selected for unstable conditions because they provide reasonable results for momentum and heat fluxes at Dome C [Vignon et al, 2016] . $L$ and thus $C_Q$ are calculated with an iterative resolution of the

Monin-Obukhov equations system. The value of $z_0$ is the mean value reported by Vignon et al [2016] for Dome C (0.56 mm). The value of $z_{0q}$ is difficult to estimate at Dome C because the very low vapor content of the atmosphere induces high uncertainties and because the scarcity of near-neutral conditions prevents an independent selection of a scheme for the stability functions. Two different approaches are used. By default, $z_{0q}=z_0$ as in King et al [2001], and in a second case, $z_{0q}$ is calculated with Andreas [1987] theoretical formula which, at Dome C, yields $z_{0q}$ values lower than $z_0$ by approximately one order of magnitude. Uncertainties on flux calculations are estimated from the variance of results obtained with the different choices of stability functions and roughness length.

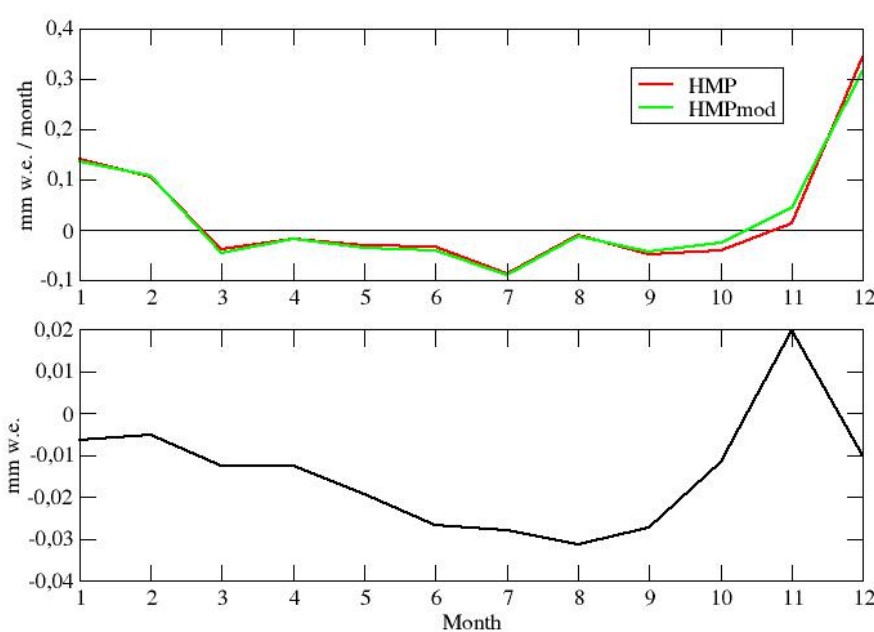

*Figure 10. Annual march of the monthly vapor flux at the surface according to HMP (red) and HMPmod( green), the black line showing 0 (upper plot), and cumulated difference (HMPmod – HMP, lower plot).*

Figure 10 shows the monthly seasonal and difference of cumulated water flux calculated by the

bulk method for 2015 using either HMP data or HMPmod reports. The flux is positive during the

summer months indicating sublimation of snow while during winter months, the flux is negative

indicating condensation to the surface.  Such seasonality is in agreement with that reported by King

et al [2001] at Halley station, coastal Antarctica but at a latitude similar to that of Dome C. The

positive summer values reflect the predominance of snow sublimation during the summer diurnal

cycle [Genthon et al., 2013] because, in summer, the surface-atmosphere exchanges are larger

during convective activity in the afternoon than in the night hours when the boundary layer

becomes stable (King et al. [2006], Vignon et al. [2016]). Integrated over the full year 2015, the net

water vapor flux is 0.2763 cm w.e. using HMPmod data and  0.2863 cm w.e using HMP data. These

numbers can vary by as much as ±100% with the different choices of stability functions and

roughness length values. They are very small anyway compared to the total surface water budget,

given that the  mean annual accumulation is about 2.5 cm w.e. [Genthon et al., 2015]. However, a

mean positive evaporation agrees with Stearns and Weidner [1993] who, for other regions of

Antarctica, conclude that the annual-mean net sublimation exceeds the annual-mean net deposition.

In fact, Figure 10 shows very little difference between calculations made with HMP and HMPmod

data: the impact of supersaturation on the water heat flux is thus very small. This is because

supersaturations predominantly occur when the wind speed and thus turbulence are weaker (figure

3) and when specific humidity is low (Figure 11), thus turbulent flux are weak (Equ. 1). A possible

contribution of blowing or drifting snow sublimation (King et al [2001], Frezzotti et al. [2004],

Barral et al. [2014]) is not taken into account in the calculations here.


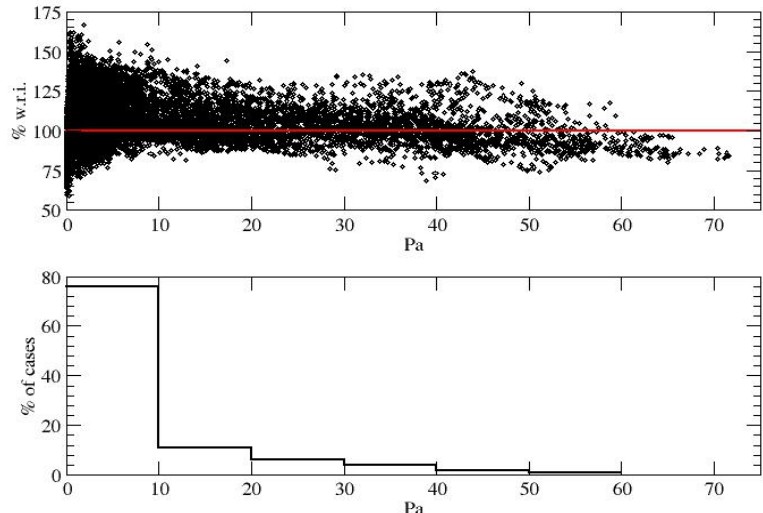

*Figure 11 : According to HMPmod, relative humidity vs water vapor partial pressure, saturation shown by the red line (upper plot), and probability distribution function of RHi above 105% with respect to water vapor partial pressure (lower plot).*


## 4. Meteorological models and cold microphysics parameterizations

The introduction (section 1) reminds the results of Genthon et al. [2013] showing that 2 models with cold microphysics parameterizations for water condensation predict significant

supersaturation at Dome C. Observations are now available to verify these results and the general models' ability to simulate the characteristics of supersaturation at Dome C. Model results from both the MAR and ECMWF are again evaluated here, although these are results from "up to date" model versions as of 2015 which somewhat differ from those in Genthon et al. [2013].

**4.1. Meteorological models and microphysics highlights**

MAR is a limited area coupled atmosphere – surface model. Atmospheric dynamics are based on

the hydrostatic approximation of the primitive equations [Gallée and Schayes, 1994]. The vertical coordinate is the normalize pressure. Near the surface where observations are made,

parameterization of turbulence in the surface boundary layer is based on the Monin-Obukhov similarity theory and turbulence above the surface boundary layer is parameterized using the K-ε model, consisting of 2 equations for turbulent kinetic energy and its dissipation. The prognostic equation of dissipation allows one to relate the mixing length to local sources of turbulence and not only to the surface. The  K-ε model used here has been adapted to neutral and stable conditions by

Duynkerke [1988]. The influence of changes in water phases on the turbulence is included following Duynkerke and Dreidonks [1987]. The relationship between the turbulent diffusion coefficient for momentum and scalars (Prandtl number) is dependent on the Richardson number following Sukoriansky et al. [2005].

Prognostic equations are used to describe 5 water species [Gallée, 1995]: specific humidity, cloud droplets and ice crystals, rain drops and snow particles. A 6th equation is added describing the number of ice crystals. Cloud microphysical parameterizations are based on Kessler [1969], Lin et al. [1983] and Levkov et al. [1992]. In particular, cloud droplets are assumed to freeze at temperatures below 238.15K while contact-freezing nucleation, deposition and condensation

freezing nucleation of ice crystals follow the formulation of Meyers et al. [1992] improved by Prenni et al. [2007]. Surface processes in MAR are modeled using the Soil-Ice-Vegetation-Atmosphere scheme (SISVAT). For the present experiment, MAR is set up over the region of Dome C with a horizontal resolution of 20 km over a 41x41 grid. Lateral forcing is taken from ERA-Interim [Dee et al., 2001]. There are 15 model levels in the vertical between the surface and 32 m,

where temperature and moisture are explicit prognostic variables of the primitive equations and parameterizations.

Numerical weather forecasts are produced by meteorological models initialized with meteorological analyses. Meteorological analyses are the result of optimally combining (assimilating)

meteorological observation from various sources (surface, radiosounding, satellites, etc) with (in) a meteorological model. Unlike observations, which are scattered in time and space, meteorological analyses have the full time and space coverage and resolution of the model. The ECMWF produces global meteorological analyses to initialize its forecasts: these are the near real-time operational analyses. Like other weather services, the ECMWF has also produced reanalyses, retrospective

analyzes for purposes other than real time operational weather forecasts. The ERA-Interim data used as lateral forcing for MAR (see above) are reanalyses produced by ECMWF. Reanalyses are more consistent in time than operational analyses because they use a same meteorological model and assimilation package while these are constantly changed towards improvement and finer resolution in the operational analyses. Some changes occurred in the ECMWF operational system in

the course of 2015 but such major aspects as horizontal and vertical resolution were not affected. Because the vertical resolution is significantly finer near the surface in the operational analyses than in the reanalyses, we elect to use here the operational analyses to compare with the observations.

The ECMWF model (versions CY40R1 and CY41R1 for the year 2015) is part of the ECMWF IFS

(Integrated Forecasting System). The ECMWF provides a full description online [2]. It is a spectral general circulation model based on the hydrostatic primitive equations. Parametrization in the surface boundary layer is again based on the Monin-Obukov similarity theory while turbulent coefficients in the unstable mixed layer above are computed using the Eddy-Diffusivity Mass Flux (EDMF) approach [Kohler et al., 2011]. They are determined above the mixed layer and in stable

conditions using a $1^{st}$ order closure based on the wind shear, a mixing length and the local Richardson number.

---

2    http://www.ecmwf.int/search/elibrary/part?solrsort=sort_label%20asc&title=part&secondary_title=40r1,
     http://www.ecmwf.int/search/elibrary/part?solrsort=sort_label%20asc&title=part&secondary_title=41r1

The cloud microphysics scheme is described in Forbes et al. [2011]. Prognostic equations are used

for cloud liquid, cloud ice, rain, and snow water contents. The scheme allows supercooled liquid

water to exist at temperatures warmer than the homogeneous nucleation threshold of 235.15K. At

temperatures colder than this, water droplets are assumed to freeze instantaneously. For

temperatures below the homogeneous freezing temperature, the scheme also assumes that ice

nucleation initiates when RHi locally reached a threshold [Karcher and Lohmann, 2002].

At the surface, the snowpack is treated taking into account its thermal insulation properties and a

representation of density [Dutra et al., 2010]. The vertical resolution in the atmosphere near the

surface is not as fine as in the MAR model. The mean elevation of the 1$^{st}$ prognostic model level at

Dome C in summer is 8.2 m, significantly higher than the observation level. Variables are also

calculated at the meteorological standard 2-m level by  interpolation between the 1$^{st}$ level and the

surface using gradient equations of the surface layer . Vignon et al. [2016] show that the surface

layer where gradient interpolation relationships are valid is often much shallower than 8 m in stable

conditions at Dome C. The 2 m interpolated values probably encompass biases due to the

interpolation formula and may have to be considered carefully. However, the elevations of the 2-m

and 1$^{st}$ level data bracket that of the observations, allowing a more detailed comparison in a region

where vertical gradients can be steep.

## 4.2. Model data comparison

Figure 12 compares the observed diurnal cycles of temperature and moisture with        the

ECMWF analyses at the 1$^{st}$ model level and at the standard 2 m level. A similar comparison is

shown with MAR at the closest model level on figure 13. There are only 4 analyses steps per day, so

ECMWF data are shown as dots on Figure 12 when the observations (48 data per day) and MAR results ((240 per day) are shown as continuous curves on figure 12 and 13.

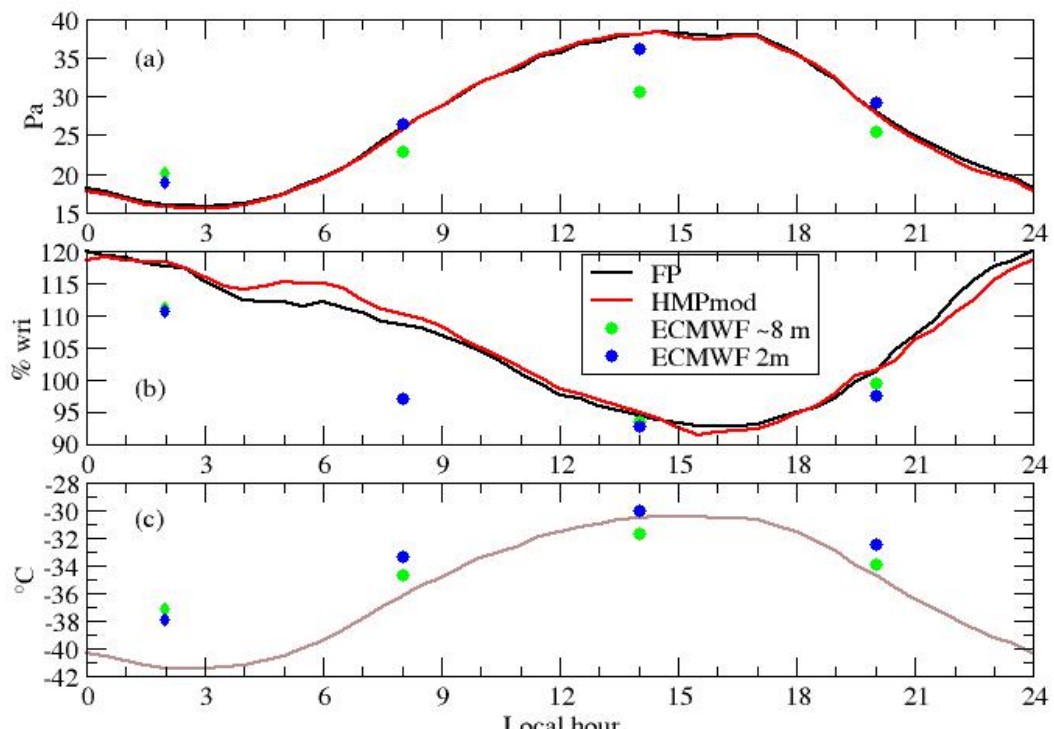

*Figure 12: Mean Dec-Jan-Feb diurnal cycle of observed (FP and HMPmod) and analysed (ECMWF, 2 m and 1$^{st}$ model level at ~8 m) water vapor partial pressure (a), relative humidity with respect to ice (b) and temperature (c). The reference temperature is that from the unmodified HMP (brown curve on plot (c)).*


The ECMWF analyses overestimate nighttime temperature and consequently underestimate the amplitude of the diurnal cycle. The amplitude of the cycle of moisture partial pressure is also underestimated but not as badly as could be expected considering a non-linear relation between temperature and saturation humidity. The model thus agrees with a large diurnal change in

magnitude and sign of the surface turbulent flux of moisture. The surface atmosphere is expectedly

moister, and the vertical gradient and turbulent flux directed upward (surface sublimation) in the early afternoon. It is downward (deposition) and much weaker at night. Because of the temperature errors, RHi is less than observed at night, yet it is significantly larger than 100%. The analyses reproduce supersaturation at night and minimum RHi in the early afternoon. MAR also produces large supersaturations, which are actually larger than the observations in summer (Figure 13a). However, the model is significantly and consistently too warm (Figure 13b), which was not the case in the model version used in Genthon et al. [2013]. Supersaturations are nonetheless a robust feature of this model.

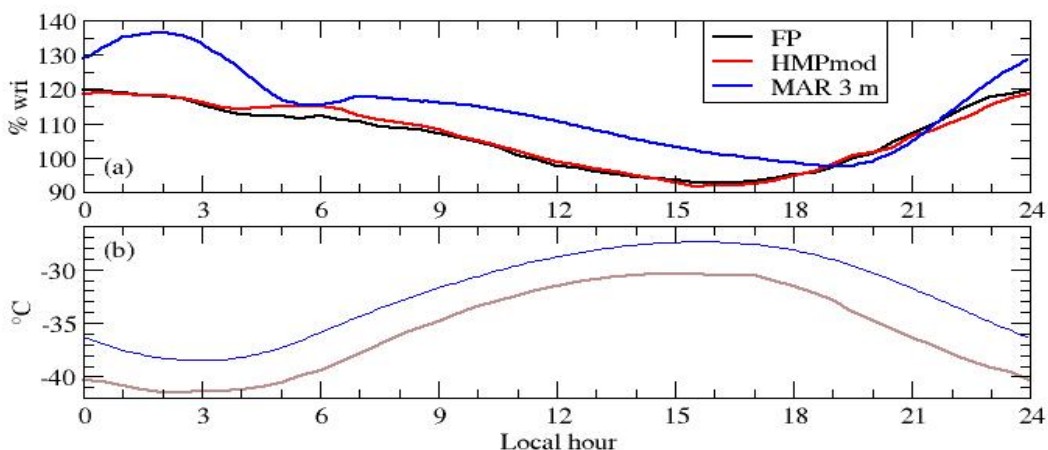

*Figure 13: Mean Dec-Jan-Feb diurnal cycle of observed (FP and HMPmod for moisture, HMP for temperature) and MAR simulated RHi (a) and air temperature (b), the brown curve being the observed as on Figure 12.*

Figure 14 displays the distribution functions of ECMWF analysed and MAR modeled RHi. This is to be compared with figure 6a for the observations. The 2 models are successful at reproducing very frequent occurrences of supersaturation, however their distributions differ both with the

observations and with each other. The MAR model is much more often supersaturated than the

observations report, and also than the ECMWF analyses.  RHi in the MAR model exceeds 200%

much more often than both the observation and the ECMWF analyzes (Not illustrated figure 14 for

scale reasons as discussed in section 3.2 / figure 6), raising particular concerns about the treatment

of the cold microphysics in this model. Differences between models and between one or the other

model and the observations are beyond observation uncertainties. Further analyses of these

differences, comparing the respective cold physics parameterizations, tracking possible

contributions of temperature biases, is beyond the scope of the present study. However, this result

illustrates that because long series of consistent in situ observations are feasible at Dome C, not

only short term chronology but also the statistics of supersaturation can be quantified and used to

exhibit differences in behavior of models and parameterizations of natural atmospheric

supersaturation.


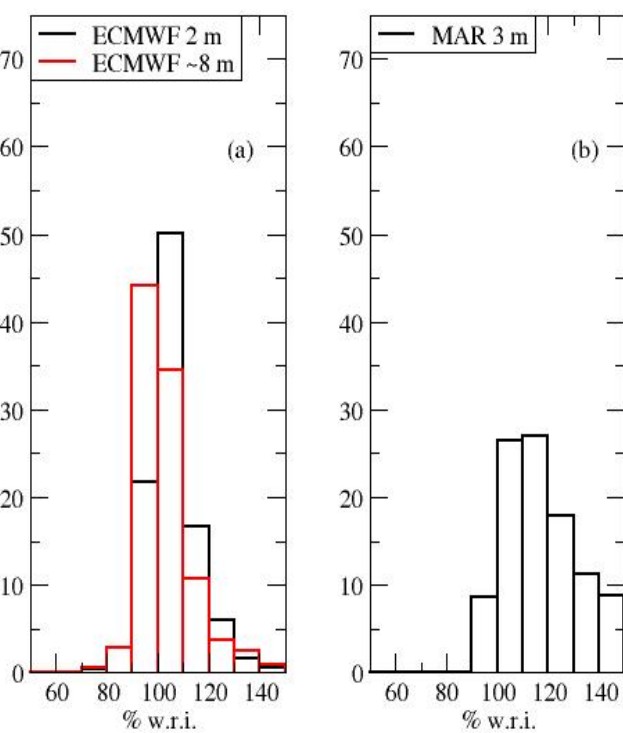

*Figure 14: ECMWF analysed (a) and MAR modeled (b) distributions of RHi in 2015 for cases of*

*RHi between 50 and 150%. The fact that a different time sampling in the models (this figure) and*

*the observations (figure 6) does not affect the comparison was verified.*

**5. Discussion and conclusions**

Major ice supersaturations are observed in the surface atmosphere of Dome C on the antarctic

plateau in atmospheric temperature and moisture conditions that are similar to those of the upper

troposphere.  To our knowledge it is the first time such strong supersaturations (up to 200%) are

observed in the natural surface atmosphere of the Earth. The presence of high ice supersaturations

suggests very low concentrations of ice nuclei (see King and Anderson [1999]). More instruments

on the tower of Dome C are being considered to detect fogs and monitor their properties, including

supercooled water fogs (see e.g. Anderson [1993]). This would help understanding the

microphysical conditions under which these ice supersaturations occur and improve microphysics

schemes used in models. Atmospheric supersaturations are frequent in the high troposphere where

cirrus clouds form [Spichtinger et al., 2003]. On the other hand, atmospheric supersaturation is an

infrequent situation in the surface atmosphere because of the high concentration of aerosols and

relatively mild temperatures which are both favorable to liquid and solid cloud formation. In this

respect, the surface atmosphere of the high antarctic plateau is a relative exception. Because of the

high albedo of snow, high latitude  and high elevation, the temperature and humidity are close to

that of the high troposphere elsewhere even in summer. Long distance transport to such remote area

is insufficient to import significant amounts of cloud and ice condensation nuclei even from the

closest sources at the oceans, thus the possibility of strong and frequent supersaturation.

Conditions for surface supersaturation may be found elsewhere in the polar regions. King and

Anderson [1999] observed supersaturations of 150% or more, and a significant frequency of ice

supersaturation of 120% or more, at the coastal antarctic station Halley. Their climatological

frequency distribution of RHi (their figure 2) has similarities with figure 6a here. This might seem

surprising as one would expect to see higher concentration of ice nuclei at a low altitude coastal site

than on the high plateau. However, the number of active ice nuclei is a strong function of

temperature [DeMott et al., 2010]. Thus it is possible that while aerosol concentrations are higher at

Halley than at Dome C, the concentration of active ice nuclei is no so much higher because of the

lower temperatures at Dome C. For the same reason, supersaturations may be expected in the

surface atmosphere of other mildly isolated polar regions but the high antarctic plateau is probably

the most propitious place for the largest and the most frequent cases of supersaturation, most similar

to that in the upper troposphere where cirrus clouds form. On the antarctic plateau itself, low

elevation clouds are a major issue for the local energy budget as even very light clouds strongly

affect the IR emissivity of the atmosphere (Gallée and Gorodetskaya [2000], Town et al. [2007]):

models that fail to reproduce supersaturation will produce too much cloudiness and fail to account

the surface energy budget.

Because they are compact, light-weight and comparatively low cost , both to buy and to operate,

solid state hygrometers (thin film capacitive sensors such as Vaisala's Humicap) are widely used to

report atmospheric moisture from radiosondes or automatic weather stations. However, these

sensors are subject to icing in supersaturated environment [Rädel and Shine, 2010] and require

correction and/or adaptation. There are not many measurements of atmospheric moisture in

Antarctica, and most (including by the radiosondes) are made using unadapted solid state sensors.

The atmospheric humidity of the antarctic atmosphere where supersaturation is frequent is likely

often underestimated from observations. Thus, the evaluation of meteorological and climate models

from these data may be biased. Observations at Dome C using modified sensors to ensure that

supersaturations can be sampled show that models that implement parameterizations of cold cloud

microphysics intended to simulate cirrus clouds at high altitude qualitatively reproduce frequent

supersaturations but fail with respect to the statistics of supersaturation events. Moreover, they fail

differently, both models tested here producing too much supersaturation but one model simulating

much more frequent occurrences of supersaturations that the other.

ECMWF and MAR supersaturation simulations are quite different for several reasons. Water vapor

concentration in the first model results from data assimilation while it is fully free to respond to

model equations and parameterizations in the second. Parameterization of ice crystal nucleation

plays a particular role in the behavior of the supersaturation process. It is based on theoretical

developments in ECMWF and in this case the number of crystals formed is rather insensitive to the

aerosol physical properties. The parameterization in MAR was developed using aircraft

observations in the Arctic. The results at Dome C probably show that parametrization tuning is too

narrow to properly account  for the near surface conditions at Dome C, although temperature

conditions probably play the most important role. Cloud ice processes are still poorly understood

and the parameterizations used here must certainly be improved. A sensitivity test of the

microphysical scheme in the RACMO meteorological model to the inclusion of supersaturation

significantly improves the performance of this model over Antarctica [van Wessem et al., 2014].

Estimations of the moisture budget of the antarctic atmosphere may  be erroneous. Because it is

comparatively undersampled by observation, studies of the antarctic atmosphere rely more than

elsewhere on models and meteorological analyses. However, only models with microphysics

parameterizations that account for supersaturation may, but not necessarily do, correctly reproduce

antarctic atmospheric moisture. Consequences of underestimating surface atmospheric moisture,

whether in observations or models not accounting for supersaturations, can include poor estimation

of precipitation, but could also be that the surface turbulent moisture exchange (evaporation or

sublimation) is erroneous. Although the ground is made of thousands of meters of snow and ice slowly accumulated through millions of years, the antarctic plateau is one of the driest places on Earth. At Dome C, only about ~30 kg m$^{-2}$ of water accumulates each year [Genthon et al. 2015]. Out

of this, the relative contribution of precipitation and evaporation is an open question. The direct measurement of both quantities is an unsolved challenge. For the turbulent fluxes, bulk and profile method parameterizations have their intrinsic limits because Monin-Obukov similarity theory requires empirical corrections functions which are not necessarily well established in very stable conditions [Vignon et al. 2016]. However, even the best theory and best parameterization deployed

based on this theory will poorly apply if the observations are wrong. The consequences are limited on the antarctic plateau though, because supersaturations are stronger and more frequent as temperature is lower, and moisture content and thus turbulent moisture flux smaller.

Finally, accurate measurements of supersaturation on the East antarctic plateau are important to

understand the physical processes involved in the water cycle in very dry conditions. In particular, it has important consequences for the formation of the isotopic signal of the snow. While the cumulated impact of water vapor exchange between the surface and the atmosphere may be small and contributes only ~10% of the surface mass balance, the asymmetry of the meteorological conditions (colder during condensation than during sublimation) leads to differences in the

fractionation coefficients for the phase transition. As supersaturation during snow accumulation induces additional fractionation [Jouzel and Merlivat, 1984], we expect a significant impact of local supersaturation to the water isotopic signal recorded in the snow [Casado et al., 2016].

Measurement of ice supersaturation as high as 200% in this very dry atmosphere invites some

revision of our understanding of the physical processes that control the water cycle in Antarctica. The deployment of more hygrometers that can measure supersaturation on the ~45-m

meteorological tower is underway and will give more insights into water vapor fluxes. Comparisons to surface observations will also improve our understanding of dry deposition and formation of frost hoar, and possibly of diamond dust. These results open new possibilities of using stations in remote

polar regions to study and understand phenomena normally occurring in clouds at several km of altitude.

**Acknowledgements:**

Support for field measurements was provided by the French polar institute IPEV through program CALVA (1013). Concordia station is jointly operated by the IPEV and PNRA. INSU provided support through programs LEFE CLAPA and DEPHY2. Support by OSUG through observatory program GLACIOCLIM is also acknowledged. The BSRN upwelling infrared radiation data which served to calculate the snow surface temperature were kindly provided by Christian Lanconelli,

CNR ISAC. The research leading to these results has received funding from the European Research Council under the European Union's Seventh Framework Programme (FP7/2007-2013) / ERC grant agreement n° [306045]. JBM also thanks UPMC university for financial assistance. We thank John King and 2 anonymous reviewers for their careful evaluation and thoughfull comments and sugestions on the initial (ACPd) version of the paper.

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
