# Peer review of "Atmospheric moisture supersaturation in the near-surface atmosphere at Dome C, antarctic plateau"

_Atmospheric Chemistry and Physics, 2016_

## Referee Comment (RC1) · J.C. King (Referee) · 7 Sep 2016

General

Making accurate measurements of atmospheric humidity in the cold, dry environment of the Antarctic plateau is challenging. This paper reports new measurements of humidity at a plateau site, Dome C, made using novel instruments that were specifically designed for accurate humidity measurement in this environment. The authors present a humidity climatology for the site and show that large supersaturations with respect to ice are frequently observed. The observations are compared with humidity simulated using both a global and a regional model and significant biases are noted in both models. The impact of the observed supersaturations on calculated surface water vapour

fluxes is examined but is found to be small when compared to climatological values of this flux.

The paper is a valuable contribution to our knowledge and understanding of near-surface atmospheric humidity over the high plateau of Antarctica. It is suitable for publication in ACP but I think that it could be improved by some restructuring. I make some suggestions on this below and list a number of other points (mostly minor) that require attention.

Major points

1.) There are three main areas of work presented in the paper: (1) comparison of different techniques for measuring humidity at Dome C, (2) presentation of a humidity climatology for Dome C and (3) use of these measurements to validate humidity in atmospheric models. At the moment, these three topics are presented partly in section 2 and partly across section 3. For example, the poor performance of the FP instrument in all but the warmest months doesn't get mentioned until section 3.2, when the year-round humidity climatology is presented. In my view, it would be more logical to first present the intercomparison of the instruments under all conditions before moving on to present the climatology and, finally, the comparison of the models with observations.

2.) In the conclusions section (lines 539-549) you state that this is the first time that ice supersaturations of up to 200% have been reported in near-surface measurements. While this may be true, King and Anderson (1999) observed occasional ice supersaturations of 150% or more, and a significant frequency of ice supersaturation of 120% or more at the coastal Antarctic station, Halley. Indeed, the climatological frequency distribution of RHice at Dome C (fig. 7a) appears quite similar to that at Halley (see King and Anderson 1999, fig. 2). This might seem surprising as one would expect to see a higher concentration of ice nuclei (IN) at a low-altitude coastal site than at Dome C and hence might expect supersaturations to be significantly lower at Halley. However, the number of active IN is a strong function of temperature (see, e.g. DeMott et al., 2010,

doi:10.1073/pnas.0910818107). Thus it is possible that, while aerosol concentrations are lower at Dome C than at Halley, the concentration of active IN at the two sites is similar because of the lower temperatures at Dome C. A comparison of supersaturation statistics for the two stations would make a useful addition to the discussion. Have any direct measurements of aerosol (particularly IN) been made at Dome C?

3.) The methods section (section 2) should include a short description of the ECMWF and MAR models with a particular focus on the representation of cloud microphysics.

4.) There is no further discussion of the biases seen in the models in the discussion and conclusions section (section 4). I appreciate that a detailed analysis of the origins of these biases is beyond the scope of the paper, but some discussion of the significance of these findings is required. It might be worth noting here that incorporating a microphysics scheme that allowed supersaturation significantly improved the performance of the RACMO model over the Antarctic (van Wessem et al, 2014, doi: 10.5194/tc-8-125-2014).

Minor points and typographical corrections

l19: analyses

l23: leaving not living

l52: condense not condensate

l58: "Conditions close to those occurring at the tropopause..."

l88: condenses not condensates

l100: King and Anderson (1999, n.b. not King et al.) report measurements at Halley, which is a coastal (not plateau) station and show that RHi is frequently between 100 and 120% at that station.

l160: condense, not condensate

l171: referred TO as A ....

l174-178: Note that the system described by K&A 1999 did heat the aspirated air.

l288: "completely sets" rather than "really sets"?

l290: Define "night" a bit more precisely?

l309: "latter two", not "latters"

l311: warmest TIME of the day

l343: night-time

l366 onwards and fig 6: These plots and discussion might fit better earlier in the section – just after fig 3, as they relate to the comparison of the instruments carried out in the first part of this section (see also major point 1).

l413-415: Some discussion of why the FP appears to sometimes give unrealistically low readings at low frost point would be useful. Possibly the air is so clean that a very high supersaturation is required to establish a frost deposit on the mirror?

l425 "at odds"

l430-435: Need to be clear that what is being described here is not a humidity climatology but a conditional climatology for vapour pressure > 2 hPa. l474: "...above thE surface..."

l476: "is written", not "writes".

l482: You should include an equation defining L.

l488: "...because OF the very low..."

l518-520 and fig 12: Maybe refer back to the equation on l469 to emphasise this point. You could replace fig 12 with two histograms showing frequency of occurrence of supersaturation as a function of (a) wind speed and (b) temperature to make your point

more clearly.

l533: "... conditions THAT ARE close ...".

l535: "in the field" – not clear if you are talking about at Dome C or elsewhere.

l535-539: "devoid of ice nuclei" may be a bit strong, but the existence of high ice super-saturations certainly suggest very low concentrations of IN (as suggested by King and Anderson, 1999). The statement about homogeneous freezing seems a little specula-tive. Is there evidence for the existence of supercooled water droplets at Dome C, e.g. from observations of rime accumulation on structures?

John King

---

## Referee Comment (RC2) · Anonymous Referee #2 · 19 Sep 2016

Review of "Atmospheric moisture supersaturation in the near-surface atmosphere at Dome C, antarctic plateau" by C. Genthon et al. MS No.: acp-2016-670

This manuscript reports surface humidity observations from Concordia station in Antarctica. It intercompares a heated humidity sensor with a frost point hygrometer and then also compares the results to models. The goal is to look at ice supersaturation. There are some comments about isotope effects and surface fluxes and how they might be affected by the results.

The paper needs major revision. The data analysis is not complete: there are high values that are eliminated and claimed to be important. I am not convinced that there may not be evaporation of ice crystals in the heated inlet, or blowing snow, leading

to anomalously high ice supersaturation measurements. There are also low biases eliminated without explanation why. I think that is because the frost point has a limited dew point, but I am not sure.

Also, the effect of ice supersaturation on isotopic fractionation is mentioned as motivation, but there is now real information here, except some passing discussion (which I do not think is correct).

Surface fluxes are also noted as an important reason for measuring near surface ice supersaturation, and some calculations are made, but these show no effect of the difference in ice supersaturation. That null result should be more prominently stated.

Detailed comments:

Page 2, L15: in general polar regions are an exception, even high latitudes of the S. ocean. It is not just Antarctica.

Page 2, L30: last sentence of abstract is awkward. Maybe state this as an implication of these results?

Page 4, L57: ice supersaturation is common at low altitudes at high latitudes, particularly in stable environments.

Page 4, L60: how close to what tropopause and when? Summer Antarctic conditions still feature mixed phase clouds and supercooled liquid to -30C (Lawson and Gettelman, 2014, PNAS). Be more specific.

Lawson, R. Paul, and Andrew Gettelman. "Impact of Antarctic mixed-phase clouds on climate." Proceedings of the National Academy of Sciences 111.51 (2014): 18156-18161.

Page 5, L84: this is a good point that highlights the uniqueness of Antarctic supersaturation.

Page 6, L112: using direct in situ measurements...

Page 7, L130: has hosted a

Page 9, L177: heating will however evaporate any ice crystals in the air, making this a total water measurement. Do you have a particle counter too? Do you know whether there are particles being evaporated? This is a critical point.

Page 12, L243: is accuracy related to the level of RH? I.e. What if it is extremely dry?

Page 15, L300: all times during the day

Page 15, L303: ,but. (Correct)

Page 15, L309: both of the latter

Page 18, L345: if the temperature and humidity do not match (the errors do not match) then is there a process problem with the ECMWF model?

Page 20, L386: what is happening for RHi > 200% ? That seems like an error. Might that affect other measurements below 200%?

Page 21, L407: has the filtering been done only on the observations? I.e. If the models produce over 150% but the observations do not, has that been reported? It should be reported .

Page 21, L414: is the difference because the frost point is too low as the air gets dry?

Page 22, L419: but there are also deviations at high moisture content. Why is that?

Page 22, L425: at odds... Reflect instrument limitations

Page 22, L435: how do you know if it is correct to remove here low points? Is this a problem with the frost point? You should know if you hit the minimum dew point.

Page 25, L453: direct estimates of

Page 29, L540: but you haven't shown them and filtered them out. Are they an error or not?

Page 31, L572: why is th flux wrong? In the previous section you have shown super-saturation does not matter for the surface fluxes. Please explain.

Page 32, L601 : most isotope schemes in models do account for kinetic effects. I don't know that this discussion of isotopes helps the manuscript very much, you only discuss it in the intro and conclusions.

Page 32, L604: again, you eliminated these from the analysis and I am not convinced they are not an error. Please show them if you are going to discuss them. What instruments showed this and how do you know it was not blowing snow/ice?
* * *

---

## Referee Comment (RC3) · Anonymous Referee #3 · 6 Oct 2016

In this paper the authors present a comprehensive analysis of supersaturation at 3m above the surface at Dome C in Antarctica. The paper formally compares hygrometers and shows the need for heated inlets otherwise, not surprisingly, frosting on the inlet maintains RH at 100. The supersaturation is then compared against two atmospheric models with microphysical schemes, which yield similar distributions for supesaturation as the observations. Finally, the authors discuss how these corrected measurements influence latent heat flux from (or to) the surface. Although this paper had a number of grammatical errors, the overall flow and logic was good. Some of the conclusions regarding the importance of supersaturation in setting the latent heat flux are not well supported. Some other factors, including the isotopic effects are simply not adequately

discussed. Overall, this paper presents new and exciting measurements and provide important insight into the approach to measuring humidity in cold and dry places. Some suggestions are listed below but otherwise following revisions this paper seems suitable for publication.

Sections 1 and 2 are very well written. For people new to this topic, the writing provides quite a good introduction. The authors do a great job explaining the pros and cons of the instruments in a way that is very practical and useful.

Larger comments: One conclusions that the authors draw is that by not considering supersaturation, estimates of moisture exchange are wrong. While this is strictly true, the oputcome of considering or not considering this effect is so small it is hardly noticeable. Latent heat exchange accounts for two order less of water accumulation than precipitation (from this manuscript) and the difference between the the HMP and HMP-mod latent heat fluxes are not really measurable. Particularly when you consider a 100% possible uncertainty from the choices of stability functions. So, I think it would be more appropriate to say that supersaturation has no measurable impact on latent heat exchange or the moisture budget at Dome C.

In my opinion the authors miss an opportunity to discuss a potentially larger impact of supersaturation, which is its effect on surface cloud formation, which greatly influences the radiation budget. So, if a model excludes superaturation, it would be a lot cloudier. As opposed to focusing on the effect of supersaturation on latent heat exchange, I would focus on the indirect effect that considering or excluding supersaturation has on the radiative budget.

There was no discussion or consideration of the uncertainty in Goff-Gatch, which could be shown by considering other alternative formulae.

Are there any aerosol sources at Dome C associated with camp activity such as diesel burning. If so, the measurements made at Dome C represents a minimum supersaturation.

In Figure 7 when comparing the observed and modeled distributions, it would be good to reduce the resolution of the observations to 4 or 6 times daily, in order to fairly compare against the models. Does the change in resolution influence the distribution.

On line 603: The authors say supersaturation at 200% is a game changer. I guess I would like to know a little more about what is implied here. What does this change?

In Figure 9 and related discussion. This looks like a kind of RH climatology, but in fact, as shown in the top panel a lot of data is excluded because Pa is too low. For this reason, I think it would better to show monthly distributions as box and whisper plots as to not present potential confusion that this is supposed to be an actual climatology.

Small comments Throughout the paper "supersaturations" is used. As noted below, I believe supersaturation should be described as a state not an event. So I would change all supersaturations to supersaturation.

13: "Superaturation….is frequent" 14: "but is very" 23 "leaving" 24: "supersaturation" 32 "can be obtained more easily at surface" 46: "against observations" 49: "in situ timeseries' " 68-70: evaporation/sublimation/condensation the role that blowing snow has on evaporation, the process is still evaporation. I think blowing snow needs to be considered zero net. 90: here and elsewhere "Antarctic" sometimes not capitalized 92: "impact the reconstruction" Here some more information on how supersaturation impacts the ice core. Is this because precip is formed under supersatration ? or is this through post deposition processes. 92: Sueprsaturation here and elsewhere should be singular. Supersaturation, in my opinion, is a process not an event. 99: "plateau seem to be capped…" 109: "such as the vertical resolution." 109: "If both models produce" 112: Maybe not "decide between models" but "diagnose the robustness of models". Ultimately, it is not the goal to remove models but to improve them all. 119: "revisited" not "reminded" 129, Again capitalize Antarctic. 137: "impact on the series" 140: "another" (not "an other") 142: extra comma after Kampfer reference 144: "were measured" 152: "2 contrasting years" 173: "energy, they have moving parts, and the

mirror..." 174: "disfunction" 201: How much is the inlet heated? Constant wattage or controlled to maintain a temperature above ambient? 202: "perform in cold" 203: "According to the manufacturer, the" 211: "for a large fraction" 238: There is a contradiction in this sentence. "up to 200%...or even more". It is either "up to" or "even more" 245: "note, that" 303: "by each instrument, but". Also, it is unclear what is meant by "reported to the atmospheric temperature of the HMP" 309: "The latter" 311: "Warmest part of the day" Figure 3: What are range or error bars on these diurnal cycles? 331: rephrase, "where classical interpolation relationships are valid". Unclear. 342 "nighttime" 346: "non-linear" 353: What is meant by: "At synoptic times" 356: "produces levels of supersaturation, which are larger than the observaions..." 381: What is meant by, "According to instrument reports..."? 384: "reaches" 385: It is unclear the justification to limit the range between 50-150%. I see that this is 99% of observations but, still, why reject 1% of the data? 396: "as expected," 412: "lose" not "loose" 421: "coldest period of" 457: "for measurements in such" 474 "the" 480: "et" to "and" 511 w.e. and also "water equivalent" is written out not as an acronym on 514 520 "associated with low" 558: "and climate models" 562: "the the" 564: "than the other" 581: "The applicability is limited to the Antarctic Plateau...."
* * *

---

## Author Comment (AC1) · 7 Dec 2016

General

Making accurate measurements of atmospheric humidity in the cold, dry environment of the Antarctic plateau is challenging. This paper reports new measurements of humidity at a plateau site, Dome C, made using novel instruments that were specifically designed for accurate humidity measurement in this environment. The authors present a humidity climatology for the site and show that large supersaturations with respect to ice are frequently observed. The observations are compared with humidity simulated using both a global and a regional model and significant biases are noted in both models. The impact of the observed supersaturations on calculated surface water vapour fluxes is examined but is found to be small when compared to climatological values of this flux.

The paper is a valuable contribution to our knowledge and understanding of near-surface atmospheric humidity over the high plateau of Antarctica. It is suitable for publication in ACP but I think that it could be improved by some restructuring. I make some suggestions on this below and list a number of other points (mostly minor) that require attention.

Major points

1.) There are three main areas of work presented in the paper: (1) comparison of different techniques for measuring humidity at Dome C, (2) presentation of a humidity climatology for Dome C and (3) use of these measurements to validate humidity in atmospheric models. At the moment, these three topics are presented partly in section 2 and partly across section 3. For example, the poor performance of the FP instrument in all but the warmest months doesn't get mentioned until section 3.2, when the year- round humidity climatology is presented. In my view, it would be more logical to first present the intercomparison of the instruments under all conditions before moving on to present the climatology and, finally, the comparison of the models with observations.

*Thank you John for raising the organization point, and for the review altogether. We certainly appreciate your suggestion of altering the flow of the paper However, it is not quite true that the FP has poor performance in all but the warmest months: the FP appears to work fine down to approximately -55°C, that is about half of the time including many days in winter. Thus I takes to presenting and discussing the winter data to raise the point. Therefore we feel necessary that, after a general presentation of the instruments (section 2), we first present summer observation which allows to asses and discard the non modified HMP, then turn to presenting the full year including winter which allows to detect the problem with the FP and limit its use in the climatology. We still feel that this is a good way to go. On the other hand, although the 3rd reviewer reports that the flow of the paper is good, we agree with you that an alternate defensible way is to push the model comparison to after a full presentation and discussion of the observations is given. We thus add a 4th section to the paper, which is fully dedicated to presenting the models including the cloud microphysics parameterizations in more details than before, then compare the models with the observations*

2.) In the conclusions section (lines 539-549) you state that this is the first time that ice supersaturations of up to 200% have been reported in near-surface measurements. While this may be true, King and Anderson (1999) observed occasional ice supersaturations of 150% or more, and a significant frequency of ice supersaturation of 120% or more at the coastal Antarctic station, Halley. Indeed, the climatological frequency distribution of RHice at Dome C (fig. 7a) appears quite similar to that at Halley (see King and Anderson 1999, fig. 2). This might seem surprising as one would expect to see a higher concentration of ice nuclei (IN) at a low-altitude coastal site than at Dome C and hence might expect supersaturations to be significantly lower at Halley. However, the number of active IN is a strong function of temperature (see, e.g. DeMott et al., 2010. Thus it is possible

that, while aerosol concentrations are lower at Dome C than at Halley, the concentration of active IN at the two sites is similar because of the lower temperatures at Dome C. A comparison of supersaturation statistics for the two stations would make a useful addition to the discussion. Have any direct measurements of aerosol (particularly IN) been made at Dome C?

*Thanks for pointing that stating that Dome C is an exception for supersaturation in the surface atmosphere is an overstatement. We have moderated the statement (thus addressing a similar comment by reviewer 2), and we now report in the discussion section your observations with Phil Anderson at Halley station, compare with ours as suggested and adopt your interpretation almost verbatim (thank you!). We do not know of observations of IN at Dome C. There are lidar observations of aerosols, which cannot see so close to the surface as our measurements, and as far as we know do not see much besides higher clouds and diamond dust, and occasional pollution plumes from the station when the wind shifts from its main origin.*

3.) The methods section (section 2) should include a short description of the ECMWF and MAR models with a particular focus on the representation of cloud microphysics.

*A quick description of the 2 models with a particular focus on the cloud microphysics is now given in the new section 4.*

4.) There is no further discussion of the biases seen in the models in the discussion and conclusions section (section 4). I appreciate that a detailed analysis of the origins of these biases is beyond the scope of the paper, but some discussion of the significance of these findings is required. It might be worth noting here that incorporating a microphysics scheme that allowed supersaturation significantly improved the performance of the RACMO model over the Antarctic (van Wessem et al, 2014, doi:10.5194/tc-8-125-2014).

*A paragraph is added in the discussion section to further address the issue of model biases, with reference to Wessem et al. This is reproduced below:*

*ECMWF and MAR supersaturation simulations are quiet different for several reasons. Water vapor*

*concentration in the model first results from data assimilation while it is fully free to respond to*

*model equations and parameterizations in the second. Parameterization of ice crystals nucleation*

*play a particular role in the behavior of the supersaturation process. It is based on theoretical*

*developments in ECMWF and in this case the number of crystals formed is rather insensitive to the*

*aerosol physical properties. It results mainly from aircraft observations in the Arctic in MAR. The*

*results at Dome C probably show that parametrization tuning is too narrow to properly account for*

*the near surface conditions at Dome C, although temperature conditions probably play the most*

*important role. Cloud ice processes are still poorly understood and the parameterizations used here*

*must certainly be improved. This point is all the more important that a sensitivity tests of the*

*RACMO microphysical scheme to the inclusion of supersaturation improves significantly the*

*performance of this model over the Antarctic [van Wessem et al., 2014].*

Minor points and typographical corrections
19: analyses

*OK, changed throughout the text*

23: leaving not living

*OK*

52: condense not condensate

*OK, changed throughout*

58: "Conditions close to those occurring at the tropopause..."

OK

88: condenses not condensates

*OK corrected throughout*

100: King and Anderson (1999, n.b. not King et al.) report measurements at Halley, which is a coastal (not plateau) station and show that RHi is frequently between 100 and 120% at that station.

*Yes, this is now reported in the discussion section*

160: condense, not condensate

*OK*

171: referred TO as A ….

*OK*

174-178: Note that the system described by K&A 1999 did heat the aspirated air.

*Yes, definitely ("Air was sampled at a nominal height of 4 m above the snow surface and taken to the unit through a 6 m long heated PTFE tube") and it was a source of inspiration. This is now clearly reported.*

288: "completely sets" rather than "really sets"?

*Yes, corrected*

290: Define "night" a bit more precisely?

*Hard to define precisely since there is no night. Now referred as "broadly the coldest half of the day".*

309: "latter two", not "latters"

*OK done*

311: warmest TIME of the day

*OK changed*

343: night-time

*OK corrected*

366 onwards and fig 6: These plots and discussion might fit better earlier in the section – just after fig 3, as they relate to the comparison of the instruments carried out in the first part of this section (see also major point 1).

*As described in the response to major point 1, after the section 2 were we present the instruments, we stick in section 3 to presenting the data along with further discussing instrument performances (including calibrating HMPmod with FP in summer because this is when both instruments perform consistently) because we need the data to assess the performances. On the other hand, the model contributions to section 3 in the initial version are now fully moved to section 4. Thus the data section along with some discussion of the instrument performance is indeed more compact than before.*

413-415: Some discussion of why the FP appears to sometimes give unrealistically low readings at low frost point would be useful. Possibly the air is so clean that a very high supersaturation is required to establish a frost deposit on the mirror?

*This occurs for low temperatures / low water content. As mentioned in section 2 when presenting the frost-point hygrometer, there in an issue with cooling the mirror to low enough temperature to reach condensation. This point is now reminded in section 3.2.*

425 "at odds"

*OK corrected*

430-435: Need to be clear that what is being described here is not a humidity climatology but a conditional climatology for vapour pressure > 2 hPa.

*OK, "When restricting to above 2 Pa" added.*

l474: "...above thE surface..."

*OK corrected*

476: "is written", not "writes".

*OK corrected*

482: You should include an equation defining L.

*we do not think that an equation of the MO length is really necessary since we refer to Vignon et al [2016] for the exact formulations of the stability functions. However, to fully make the point, we also add the reference to Stull [1990] from whom the formulation of L is obtained.*

488: "...because OF the very low..."

*This does not sound right because the sentence is "... because the very low vapor content of the atmosphere induces high uncertainties...". Should we write "because of the very low vapor content, high uncertainties are induced?"*

518-520 and fig 12: Maybe refer back to the equation on l469 to emphasize this point. You could replace fig 12 with two histograms showing frequency of occurrence of supersaturation as a function of (a) wind speed and (b) temperature to make your point more clearly.

*Reference to the equation is now given. Figure 3 was introduced in response to a comment by reviewer 2, showing RHi as a function of wind speed. Reference to new figure 3 is given here, answering part of the request for additional figures. As figure 3 illustrates the point for wind speed, we do not feel that the same figure for IR would add much to the discussion and this is omitted.*

533: "... conditions THAT ARE close ...".

*OK, changed*

535: "in the field" – not clear if you are talking about at Dome C or elsewhere.

*Yes, at Dome C, clarified*

535-539: "devoid of ice nuclei" may be a bit strong, but the existence of high ice super- saturations certainly suggest very low concentrations of IN (as suggested by King and Anderson, 1999). The statement about homogeneous freezing seems a little speculative. Is there evidence for the existence of supercooled water droplets at Dome C, e.g. from observations of rime accumulation on structures?

*There are evidences of liquid water in clouds aloft but no evidence of supercooled water droplets near the surface. Extensive dendricity clearly indicates that frost deposition is from inverse sublimation rather than riming. The text is changed to moderate the statement of "devoid ice nuclei", citing Anderson [1993].*

*We thank John King for his thoughtful comments and suggestions which have definitely contributed enhancing the quality of the paper.*

---

## Author Comment (AC2) · 7 Dec 2016

Review of "Atmospheric moisture supersaturation in the near-surface atmosphere at Dome C, antarctic plateau" by C. Genthon et al. MS No.: acp-2016-670 This manuscript reports surface humidity observations from Concordia station in Antarctica. It intercompares a heated humidity sensor with a frost point hygrometer and then also compares the results to models. The goal is to look at ice supersaturation. There are some comments about isotope effects and surface fluxes and how they might be affected by the results. The paper needs major revision. The data analysis is not complete: there are high values that are eliminated and claimed to be important. I am not convinced that there may not be evaporation of ice crystals in the heated inlet, or blowing snow, leading to anomalously high ice supersaturation measurements. There are also low biases eliminated without explanation why. I think that is because the frost point has a limited dew point, but I am not sure. Also, the effect of ice supersaturation on isotopic fractionation is mentioned as motivation, but there is now real information here, except some passing discussion (which I do not think is correct).

Surface fluxes are also noted as an important reason for measuring near surface ice supersaturation, and some calculations are made, but these show no effect of the difference in ice supersaturation. That null result should be more prominently stated.

*Replies and accounting of general comments:*

*1) High values are not merely eliminated (they are mentioned in the text) but they were not initially illustrated on figures because occurrence is comparatively rare and displaying requires using logarithmic vertical axis which somewhat distorts the overall figure rendering. To answer the reviewer's comment while keeping readable figures, the distributions both to 150 and to 200% are now shown on figure 5, the later with a logarithmic scale. This is reproduced below.*

[Figure]

*Figure 1: Observed distribution of Rhi in 2015*

*2) The reviewer raises an important concern which we have admittedly omitted: that some of the supersaturations we measure may be artifacts resulting from the evaporation of solid particles in*

*the heated inlet. This is a major issue and we do not have measurements of ice particles at the level of the instrument to directly rule this out. However, if supersaturation results from evaporation of blowing snow particles, occurrence of supersaturation should correlate with wind speed as blowing snow only occurs with higher wind speed. Figure 2 below shows that supersaturation actually anticorrelates with wind speed. Strong winds which can erode snow from the surface are associated with the intrusion of oceanic air masses carrying comparatively larger quantities of aerosols preventing supersaturation, thus probably the anticorrelation.*

[Figure]

*Figure 2: scatter plot of measured Rhi versus measured 10-m wind speed.*

*Concerning pecipitation particles, even diamond dust strongly affects the sky downwelling IR radiation (Town et al, 2006, Cloud cover over south pole from visual observation, satellite retrieval, and surface based infrared radiation measurement, J. Clim., 20, 544-559; Galllée and Gorodetskaya, 2010, see paper liste of references). Then again, if solid particles from precipitation bias our reports of supersauration, supersaturation should correlate with IR. Figure 3 below shows the opposite. More supersaturation when IR decreases probably largely reflects that relative humidity increases if total humidity is conserved while the atmosphere radiatively cools.*

[Figure]

*Figure 3: scatter plot of measured Rhi versus measured downward longwave radiation.*

*The point raised by the reviewer is nonetheless very important and is now discussed in section 2, and although only one of the 2 figures above is shown for the sake of conciseness the 2 anticorrelations in support of a limited impact are now reported.*

*3) Concerning the unrealistically low values, the reviewer is correct, they are eliminated because they result from the hygrometer having a limited frost point temperature. This limit was initially mentioned in section 2 presenting the instruments and methods but not mirrored in section 3.2 where the unrealistically low values are identified. This is now corrected.*

*4) Isotopes are admittedly not a major motivation for the present study. It remains that supersaturations may have consequences on the use of water isotopes to interpret ice core recors, a point which we think worth reporting although we do not address it really. As the other reviewers do not complain about our mentioning the water isotopes issue we keep in an abridged for droping all numerics and formula. The reviewer does not detail why he/she "does not think the discussion is correct". We hope the new discussion also resolves this issue.*

*5) It is not true that ignoring supersaturations has no effect on calculations of sublimation. However,the effect is small and this is prominently stated in the abstract: "This is unlikely to strongly affect estimations of sublimation".*

Detailed comments:
Page 2, L15: in general polar regions are an exception, even high latitudes of the S. ocean. It is not just Antarctica.

*OK. "An exception" is replaced by "one exception" and In response to reviewer 1 we now discuss observations in coastal Antarctica that also report supersaturations. We thus moderate our statement and report that supersaturations are observed at other places in polar regions.*

Page 2, L30: last sentence of abstract is awkward. Maybe state this as an implication of these results?

*OK, the sentence is reformulated. This is an implication of the fact that we demonstrate that supersaturations are indeed strong and very frequent at the surface at Dome C, and they can be comparatively easily measured providing extensive samples to test models.*

Page 4, L57: ice supersaturation is common at low altitudes at high latitudes, particularly in stable environments.

*Although we are not convinced that ice supersaturation is "common" (there is not much litterature that highlights the fact), it certainly occurs at in the high latitudes at low elevation. We have moderated our statement and now report that the antarctic high plateau is "relative" exception. We cite King and Anderson [1999] who report a distribution of relative humidity with significant occurrences of supersaturation at an antarctic station near the coast, and compare our distribution with theirs. We adopt the interpretation by the 1st reviewer that, although there are more ice nuclei at the coast, they may be less active than at Dome C because of the warmer temperature and thus allow significant supersaturation.*

Page 4, L60: how close to what tropopause and when? Summer Antarctic conditions still feature mixed phase clouds and supercooled liquid to -30C (Lawson and Gettelman, 2014, PNAS). Be more specific. Lawson, R. Paul, and Andrew Gettelman. "Impact of Antarctic mixed-phase clouds on climate." Proceedings of the National Academy of Sciences 111.51 (2014): 18156- 18161.

*This part reads: "Conditions close to the"tropopause are however found over the antarctic ice sheet both in terms of temperature and humidity levels. Because of the distance from the nearest coasts and the high elevation, the antarctic plateau is also particularly secluded from sources of aerosols". We believe this says it all: there are similarities not only with respect to temperature and humidity but also isolation from sources of impurities (although admittedly not everywhere, as deep and frequent convection brings impurities from the surface in the tropics). The point is that we are discussing a surface atmosphere here, where in situ observations are easier than he upper trosphere and which directly iteracts with the surface. We are not sure here what is the reviewer's point concerning the mixed phase clouds.*

Page 5, L84: this is a good point that highlights the uniqueness of Antarctic supersaturation.

*OK*

Page 6, L112: using direct in situ measurements…

OK

Page 7, L130: has hosted a

OK

Page 9, L177: heating will however evaporate any ice crystals in the air, making this a total water measurement. Do you have a particle counter too? Do you know whether there are particles being evaporated? This is a critical point.

*Right, this is a critical point. We do not have a particle counter but we provide arguments above (also now in the paper) in support of a limited impact if any.*

Page 12, L243: is accuracy related to the level of RH? I.e. What if it is extremely dry?

The data sheet ([http://www.vaisala.fi/Vaisala%20Documents/Brochures%20and%20Datasheets/HMP155-Datasheet-B210752EN-E-LoRes.pdf](http://www.vaisala.fi/Vaisala%20Documents/Brochures%20and%20Datasheets/HMP155-Datasheet-B210752EN-E-LoRes.pdf)) specifies different accuracy for different humidity range only for the warmer temperatures. It may be assumed that at cold temperature, the air is quite dry anyway and the main sensitivity is to temperature.

Page 15, L300: all times during the day

OK

Page 15, L303: ,but. (Correct)

OK

Page 15, L309: both of the latter

replaced by "the latter 2" following an other reviewer.

Page 18, L345: if the temperature and humidity do not match (the errors do not match) then is there a process problem with the ECMWF model?

Not necessarily. The relation of humidity with temperature is a complex one both in the real world

and in a model, much less direct and much more non linear than a mere Clausius – Clapeyron.

Page 20, L386: what is happening for RHi > 200% ? That seems like an error. Might that affect other measurements below 200%?

We do not think the 200% is an error, but because they are infrequent, it did not seem practical to show the distributions to the extent of the 200% occurrences. This is now done by showing the the distributions both with a linear and a log scale for relative frequency of occurrence (figure 6).

Page 21, L407: has the filtering been done only on the observations? I.e. If the models produce over 150% but the observations do not, has that been reported? It should be reported .

Yes, both models can produce over 150% saturation although the MAR model does that much more (too) frequently. This is now reported, particularly raisong the issue for the MAR model.

Page 21, L414: is the difference because the frost point is too low as the air gets dry?

Yes. The issue is raised when presenting the instruments in section 2, it is now reminded here.

Page 22, L419: but there are also deviations at high moisture content. Why is that?

Deviations are quite small at high moisture content. Some deviations show at intermediate moisture content and we do not know why. It is only in the dryest cases that the deviations kill any significant regression.

Page 22, L425: at odds... Reflect instrument limitations

OK

Page 22, L435: how do you know if it is correct to remove here low points? Is this a problem with the frost point? You should know if you hit the minimum dew point.

This most probably is a problem with the FP as the other hygrometer and the models do not report values any close to these. The FP needs to cool a mirror using a Peltier cell. There is a limit to the cooling a Peltier cell can provide considering how efficient heat extraction is. The hygrometer documentation loosely refers to a -65°C operational limit which we suspect is related to the theoretical ability to cool the mirror down to -65°C but then if the real relative humidity less than 100% the limit can be reached at ambient temperature above -65°C.

Page 25, L453: direct estimates of

OK

Page 29, L540: but you haven't shown them and filtered them out. Are they an error or not?

They are now shown figure 6.

Page 31, L572: why is the flux wrong? In the previous section you have shown super-saturation does not matter for the surface fluxes. Please explain.

"Wrong" is a bit too strong, this is now replaced by "erroneous". Even though this makes little

difference for the annual integrated turbulent moisture exchange, this results from alternation of frost deposition and sublimation, particularly in summer when sublimation during the day alternates with deposition during the night. At such time scale the issue may be more important.

Page 32, L601 : most isotope schemes in models do account for kinetic effects. I don't know that this discussion of isotopes helps the manuscript very much, you only discuss it in the intro and conclusions.

Isotope models allow for kinetic effects but kinetic effects related to supersaturation can only proceed if supersaturation is there. However, we agree with the reviewer, in the previous version we wrote either too much or to little about water isotopes. We do think it is important to highlight a potential issue for isotopes, but we now reduce this to men,tioning it in the introduction and a quick reminder in the discussion.

Page 32, L604: again, you eliminated these from the analysis and I am not convinced they are not an error. Please show them if you are going to discuss them. What instruments showed this and how do you know it was not blowing snow/ice?

We do show them now on figure 6. We do now provide some evidence that ice particles are not a major issue with our measurements. We nonetheless agree that particularly with the most infrequent extreme values such as 200% RHi an ice particle artifact cannot be ruled out. This is now reported in the new text.

We thank the reviewer for his/her thoughtful remarks and suggestions, particularly with raising the ice particles issue which definitely needs to be acknowledged and adressed.

---

## Author Comment (AC3) · 7 Dec 2016

In this paper the authors present a comprehensive analysis of supersaturation at 3m above the surface at Dome C in Antarctica. The paper formally compares hygrometers and shows the need for heated inlets otherwise, not surprisingly, frosting on the inlet maintains RH at 100. The supersaturation is then compared against two atmospheric models with microphysical schemes, which yield similar distributions for supersaturation as the observations. Finally, the authors discuss how these corrected measurements influence latent heat flux from (or to) the surface. Although this paper had a number of grammatical errors, the overall flow and logic was good. Some of the conclusions regarding the importance of supersaturation in setting the latent heat flux are not well supported. Some other factors, including the isotopic effects are simply not adequately discussed.

Overall, this paper presents new and exciting measurements and provide important insight into the approach to measuring humidity in cold and dry places. Some suggestions are listed below but otherwise following revisions this paper seems suitable for publication. Sections 1 and 2 are very well written. For people new to this topic, the writing provides quite a good introduction. The authors do a great job explaining the pros and cons of the instruments in a way that is very practical and useful.

Larger comments: One conclusions that the authors draw is that by not considering supersaturation, estimates of moisture exchange are wrong. While this is strictly true, the outcome of considering or not considering this effect is so small it is hardly noticeable. Latent heat exchange accounts for two order less of water accumulation than precipitation (from this manuscript) and the difference between the the HMP and HMP- mod latent heat fluxes are not really measurable. Particularly when you consider a 100% possible uncertainty from the choices of stability functions. So, I think it would be more appropriate to say that supersaturation has no measurable impact on latent heat exchange or the moisture budget at Dome C. In my opinion the authors miss an opportunity to discuss a potentially larger impact of supersaturation, which is its effect on surface cloud formation, which greatly influences the radiation budget. So, if a model excludes supersaturation, it would be a lot cloudier. As opposed to focusing on the effect of supersaturation on latent heat exchange, I would focus on the indirect effect that considering or excluding supersaturation has on the radiative budget. There was no discussion or consideration of the uncertainty in Goff-Gatch, which could be shown by considering other alternative formulae. Are there any aerosol sources at Dome C associated with camp activity such as diesel burning. If so, the measurements made at Dome C represents a minimum supersaturation.

*Replies and accounting of the larger comments:*

*We agree that initially concluding about the turbulent moisture flux that "the consequences (of occurrences of supersaturations) are limited on the antarctic plateau" is an understatement. We replace with "no measurable impact".*

*We also agree that occurrences and properly accounting in models of supersaturations have a strong impact on clouds and radiation – this is actually how we make our introduction in the topic, raising the issue of parameterization of high (cirrus) clouds that strongly affect the Earth energy budget. We did not raise the local cloud issue though. This is corrected in the conclusion but we do not attempt to estimate the impact on the radiation as this is much less straightforward than estimating turbulent fluxes: there is not bulk approach like Monim-Obukov formulations available, one would need to run models of both cloud formation and radiation transfer.*

*Finally, yes we also agree that there are uncertainties in existing empirical formulas to calculate saturation vapor pressure and thus supersaturations. As temperatures get colder, the differences between the formulas get larger (see e.g. http://cires1.colorado.edu/~voemel/vp.html). It is interesting that alternative formulations are generally evaluated against Goff-Gratch formulaes:*

*those are widely considered a reference mile stone. When mentioning that we use Goff and Gratch, we now also report that other formulas exist and that they can result in differences of up to 20% in the estimation of the saturation water vapor and thus on supersaturation for the coldest temperatures – which are alos the least frequent. Note that fro the HMP, from which humidity for the coldest cases are obteined, the conversion formulaees actually convert from RH with respect to liquid to RH with respect to solid, with a 5° (heating) correction, so this is largely a 2 way correction though which potential errors partially cancel.*

*Replies to the more detailed comments:*

In Figure 7 when comparing the observed and modeled distributions, it would be good to reduce the resolution of the observations to 4 or 6 times daily, in order to fairly compare against the models. Does the change in resolution influence the distribution.

*The figure below shows little difference whether all available observations or only observations at the synoptic times (0, 6, 12, 18 h TU, the sampling of the ECMWF data) are used to build the distributions. This is now reported in the the legend on figure 9, mentioning that the comparison between the model and observations is not affected. Because differences are so small, we prefer showing the distribution from the full time resolution of the observations and the models., reflecting the full information in the available data.*

[Figure]

*Figure: Distributions of Rhi from observations, horiginal (30') time sampling (left) and 6-hour subsampling (right)*

On line 603: The authors say supersaturation at 200% is a game changer. I guess I would like to know a little more about what is implied here. What does this change?

*"Game changer" is probably a little beyond our though. We change this to "invites some revision of our understanding...", considering in particular the issues raised concerning water isotopes.*

In Figure 9 and related discussion. This looks like a kind of RH climatology, but in fact, as shown in the top panel a lot of data is excluded because Pa is too low. For this reason, I think it would better to show monthly distributions as box and whisper plots as to not present potential confusion that this is supposed to be an actual climatology.

*Following comments by reviewer 2, we have straightened the discussion on the fact that we exclude*

*extreme (and extremely unlikely) values, so it is now clearer that the climatology is restricted to a specific range of values. We suppose that the reviewer refers to "box and whisker" rather than "box and whisper". It is not clear how box and whisker helps with a potential confusion with an actual climatology : showing plots for 3 instruments with and without restriction, we think that box and whisker actually add confusion to the figure.*

Small comments Throughout the paper "supersaturations" is used. As noted below, I believe supersaturation should be described as a state not an event. So I would change all supersaturations to supersaturation.

*We appreciate the linguistic case made by the reviewer. However, we do find "supersaturations" used in various publications including at least 7 in ACP. Sticking to using "supersaturations", is possibly linguistically improper but scientifically convenient: it allows to indeed distinguish between the state (without s) and the recorded events (with s).*

13: "Superaturation. . ..is frequent"

*And yes, here we agree that "supersaturation" is more correct*

14: "but is very"

*OK*

23 "leaving"

*OK*

24: "supersaturation"

*OK here too*

32 "can be obtained more easily at surface"

*This part of the abstract was reformulated following comment by an other reviewer*

46: "against observations"

*OK*

49: "in situ timeseries' "

*OK, replaced by "in situ observation timeseries"*

68-70: evaporation/sublimation/condensation the role that blowing snow has on evaporation, the process is still evaporation. I think blowing snow needs to be considered zero net.

*The distinction between surface and blowing snow evaporation  is generally made because, from a calculation and modeling perspective, these are different processes. We calculate surface evaporation in section 3.3 using Monin-Obukov similarity theory. This cannot apply to blowing snow evaporation (and we mention that blowing snow is not considered but is infrequent anyway).*

90: here and elsewhere "Antarctic" sometimes not capitalized

*Antarcica, as a region name, is capitalized, antarctic, as a adjective, is not.*

92: "impact the reconstruction" Here some more information on how supersaturation impacts the ice core. Is this because precip is formed under supersaturation ? or is this through post deposition processes.

*The impact is through kinetic effect each time the phase changes. This is mentioned in ther introduction ("Supersaturation leads to kinetic fractionation of the stable isotopic composition of water when it condenses") Although the net budget is small, water is constantly exchanged between the snow surface and the atmosphere in one direction or the other. The cumulated impact on water isotopes may not be negligible. We are lighter on the isotopic issue in the present version, removing all equations, however this particular issues is now highlighted in the discussion section: " it has important consequences for the formation of the isotopic signal of the snow. While the cumulated impact of water vapor exchange between the surface and the atmosphere may be small and contributes only ~10% of the surface mass balance, the asymmetry of the meteorological conditions (colder during condensation than during sublimation) leads to differences in the fractionation coefficients for the phase transition. As supersaturation during snow accumulation induces additional fractionation [Jouzel and Merlivat, 1984], we expect a significant impact of local supersaturation to the water isotopic signal recorded in the snow [Casado et al., 2016]".*

92: Supersaturation here and elsewhere should be singular. Supersaturation, in my opinion, is a process not an event.

*Please see above*

99: "plateau seem to be capped. . ."

*We stick to the formulation "reach but seem to be capped" because it also conveys the additional information that the unmodified sensors can measure 100% relative humidity even in the Dome C conditions.*

109: "such as the vertical resolution."

*OK done*

109: "If both models produce"

*OK done*

112: Maybe not "decide between models" but  diagnose the robustness of models". Ultimately, it is not the goal to remove models but to improve them all.

*Finding that one model does better than the other hints at a better approach to parameterize supersaturation. In that sense it is really "decide between models" just like in a model intercomparison exercise. However, it is true that the ultimate goal is to improve our modeling capabilities. Therefore, we change this to "decide between and improve models".*

119: "revisited" not "reminded"

129, Again capitalize Antarctic.

*Please see above*

137: "impact on the series"

*OK*

140: "another" (not "an other")

*OK*

142: extra comma after Kampfer reference

*OK*

144: "were measured"

*Actually, they are still being measured so we stick to "are measured"*

152: "2 contrasting years"

*OK*

173: "energy, they have moving parts, and the mirror. . ."

*They have moving part **because** the mirror needs to be cleaned. This is now clarified, using "because" rather than "as"*

174: "disfunction"

*Both dis and dys appear possible but OK for disfunction*

201: How much is the inlet heated? Constant wattage or controlled to maintain a temperature above ambient?

*This is a good question and is not mentioned in the manual. As we report in the paper, we never see any frost deposition so this is bound to be above saturation temperature.*

202: "perform in cold"

*OK*

203: "According to the manufacturer, the"

*OK*

211: "for a large fraction"

*OK*

238: There is a contradiction in this sentence. "up to 200%...or even more". It is either "up to" or

"even more"

*This part is reformulated following an other reviewer's comment*

245: "note, that"

*OK*

303: "by each instrument, but". Also, it is unclear what is meant by "reported to the atmospheric temperature of the HMP"

*This is reformulated: "...reported to one same atmospheric temperature, that of the (unmodified)*

*HMP..."*

309: "The latter"

*OK*

311: "Warmest part of the day" Figure 3: What are range or error bars on these diurnal cycles?

*Because in summer all instruments perform within their nominal temperature range, the errors are the instrumental errors provided by the manufacturers as stated in section 2. This is very small compared to the amplitude of the diurnal cycle, is almost the same at all times, and is not considered essential for this figure.*

331: rephrase, "where classical interpolation relationships are valid". Unclear.

*OK "classical" replaced with "gradient".*

342 "nighttime"

*OK*

346: "non-linear"

*OK*

353: What is meant by: "At synoptic times"

*Right, this is meteorologist jargon but is now removed in the new text.*

356: "produces levels of supersaturation, which are larger than the observations. . ."

*OK*

381: What is meant by, "According to instrument reports. . ."?

*This was to distinguish between observation and model data, but following reviewer's 1 comment, the models no longer show in this part (they are moved to a specific model section 4) resolving any possible ambiguity. Thus "According to instruments reports" is simply removed.*

384: "reaches"

*OK*

385: It is unclear the justification to limit the range between 50-150%. I see that this is 99% of observations but, still, why reject 1% of the data?

*The justification for rejecting the most extreme data from the figure was that they are infrequent and using a linear vertical axis they would not clearly come out of the axis line itself. To resolve this problem, we now show 2 graphs side by side (new fig 6), one with a linear vertical axis restricted to 50-150% RHi, the other with a log vertical axis extending to 200%.*

396: "as expected,"

*OK*

412: "lose" not "loose"

*OK*

421: "coldest period of"

*OK (periods rather than period)*

457: "for measurements in such"

*OK*

474 "the"

*OK*

480: "et" to "and"

*OK*

511 w.e. and also "water equivalent" is written out not as an acronym on

*OK*

514 520 "associated with low"

*This is reformulated*

558: "and climate models"

*OK*

562: "the the"

*OK*

564: "than the other"

*OK*

581: "The applicability is limited to the Antarctic Plateau. . .."

*We don't know whether this is a comment or a request for changing something in the text? We take this as a comment and, yes, this is limited to the conditions found on the antarctic plateau.*

*We are grateful to the reviewer for his/her very careful reading of the paper, many comments and suggestions which have definitely contributed produce a better paper.*